# Dynamical development of strength and stability of asteroid material under 440 GeV proton beam irradiation

M. Bochmann [1] ✉, K.-G. Schlesinger[1], C. D. Arrowsmith [2,3], P. Alexaki[4,5], M. Alfonso Poza[4], M. Ambarki[4], E. M. Andersen[4,6], P. J. Bilbao [7], R. Bingham [8,9], F. D. Cruz[7], A. Ebn Rahmoun[4], A. M. Goillot[4], J. W. D. Halliday [2,8], B. T. Huffman[2], E. Kamenicka [4], M. Lazzaroni[4], B. Lloyd[2], E. E. Los [2,10], J.-M. Quetsch[4], B. Reville [11], P. Rousiadou[12], S. Sarkar [2], L. O. Silva [7], P. Simon [13], E. Soria[4], V. Stergiou[2,4], S. Zhang[2], N. Charitonidis [4] & G. Gregori [2]

Asteroid materials experience rapid thermoelastic and plastic stress evolution when subjected to high-energy irradiation – an effect that has not previously been captured through non-destructive, time-resolved experiments. Yet, accurate modeling of asteroid deflection scenarios, such as those proposed for planetary defense, critically depends on precise knowledge of the material's mechanical behavior under extreme conditions to predict kinetic energy transfer and orbital deviation. In an experimental campaign at CERN's High Radiation to Materials facility (HiRadMat), we irradiated a Campo del Cielo iron meteorite sample with 440 GeV protons from the Super Proton Synchrotron. Using Laser Doppler Vibrometry, we captured the resulting thermally induced stress waves in real time. Our results demonstrate that asteroid materials can absorb significantly more energy without structural failure than normal material parameters would suggest. Crucially, we were able to reproduce – under controlled laboratory conditions – the discrepancy factor observed between laboratory-derived yield strength values and those inferred from atmospheric meteor breakup events.

Planetary defense has been a topic of growing scientific and strategic interest over the past decades, with a range of mitigation concepts proposed, including kinetic impactors and stand-off nuclear explosions[1]. While numerous simulation studies have modeled asteroid deflection scenarios under such conditions, the reliability of these models critically depends on an accurate understanding of the material response of asteroid constituents under extreme energy deposition.

Recent missions such as NASA's Double Asteroid Redirection Test (DART) mission have demonstrated the feasibility of kinetic impactor strategies[2]. However, experimental data on the rapid thermoelastic and plastic stress eveolution of asteroid material under high-energy

[1]BoS GmbH/OuSoCo, Mörbisch am See, Austria. [2]Department of Physics, University of Oxford, Oxford, UK. [3]University of Rochester Laboratory for Laser Energetics, Rochester, NY, USA. [4]European Organization for Nuclear Research (CERN), Geneva, Switzerland. [5]Department of Physics, National Kapodistrian University of Athens, Athens, Greece. [6]University of Bergen, Bergen, Norway. [7]GoLP/Instituto de Plasmas e Fusão Nuclear, Instituto Superior Técnico, Universidade de Lisboa, Lisboa, Portugal. [8]STFC & Department of Physics, University of Strathclyde, Glasgow, UK. [9]Rutherford Appleton Laboratory, Didcot, Oxfordshire, UK. [10]Department of Physics, Imperial College London, London, UK. [11]Max-Planck-Institut für Kernphysik, Heidelberg, Germany. [12]Department of Physics, University of Ioannina, Ioannina, Greece. [13]GSI Helmholtzzentrum für Schwerionenforschung GmbH, Darmstadt, Germany. ✉e-mail: melanie@bos-gmbh.io

irradiation remains extremely scarce. Accurate modeling of asteroid material response under extreme conditions necessitates overcoming two principal challenges: (1) the lack of real-time data on the evolution of mechanical properties under irradiation, and (2) a significant discrepancy between laboratory-based measurements and yield strengths inferred from atmospheric meteor breakups. For instance, nanoindentation studies (e.g., Ueki et al.[3]) often yield values up to a factor of seven higher than those derived from ram-pressure models[3,4]. Mulford et al.[5] hypothesized that meteorites may exhibit mechanical behavior akin to that of complex composite materials. Smirnov and Konstantinov[6] demonstrated that the strain-rate dependence of yield strength in complex materials can be attributed to internal structural dynamics. A similar mechanism could account for the behavior observed in meteorites, including the pronounced discrepancy between yield strength values obtained from nanoindentation and those inferred from ram-pressure models[3,4].

Iron meteorites, which originate from metal-rich asteroids, offer a unique opportunity to investigate these effects. While various studies have characterized their mechanical properties, most were conducted on cold or unaltered samples[7]. Studies like that of Jain et al.[8] analyzed stress signatures from ancient asteroid collisions using 119 meteorite specimens, and Siraj et al.[4] inferred strength parameters from atmospheric breakup events. However, neither study captured the real-time evolution of material properties under energetic impact.

Laboratory efforts have also attempted to emulate impact conditions. For example, the Z-pinch pulsed power facility at Sandia National Laboratories was used to irradiate meteorite samples with soft X-rays, simulating nuclear-sourced surface energy deposition[9,10]. Yet, the destructive nature of these tests precluded direct measurement of the resulting material response. More recently, Moore et al.[11] demonstrated the complete momentum transfer onto scaled asteroid targets in laboratory conditions. While groundbreaking, their setup again did not permit tracking material deformation or yield strength evolution, despite the known dependence of momentum transfer efficiency on the material's internal state.

In this study, we present an experiment conducted at the High-Radiation to Materials (HiRadMat) facility of CERN[12], in which a meteorite sample was exposed to a high-intensity, high-energy proton beam. The dynamic response of the sample, including momentum transfer and elastic/plastic behavior, was recorded in real time via non-destructive methods. To our knowledge, this is the only laboratory experiment to measure the real-time evolution of yield strength and damping behavior of asteroid-representative material under high-energy irradiation. The experiment was developed in collaboration with OuSoCo (Outer Solar System Company), which is developing a proprietary method for generating high-energy proton beams in space, with beam properties comparable to those of CERN's Super Proton Synchrotron (SPS). While the technical details of in-space beam generation fall outside the scope of this paper, the results presented here are, in principle, transferable to any deflection method where the applied energy penetrates deeply into the target material. We focus this study on metal-rich asteroids, whose relative material homogeneity facilitates characterization and modeling. However, since stony asteroids represent the most common class of near-Earth objects, future experimental campaigns are already planned to extend this analysis to silicate-rich meteorite samples.

## Results

It is common for meteorite samples to be composed of multiple phases of iron-nickel alloy. The utilized fragment (imaged in Fig. 1) is taken from the Campo del Cielo iron meteorite and it features a characteristic two-phase crystal structure consisting of Kamacite, i.e., ferritic iron ($\alpha$ structure), and Taenite, i.e., Austenite ($\gamma$ structure)[13]. Once cut into a cylindrical shape, the phase boundaries become clearer (not to be confused with cracks). Scanning electron microscopy (SEM) and energy-dispersive X-ray spectroscopy reveal clearly visible phase boundaries (pictures are provided in Supplementary Information).

The dynamical response and development of yield strength of asteroid material are examined experimentally by irradiating an iron meteorite sample with beams of high-energy 440 GeV protons extracted from CERN's SPS. When such high-energy protons collide with atomic nuclei in the sample, hadronization of quarks and gluons leads to the generation of hadronic and electromagnetic cascades of secondary particles. These include (but are not limited to) pions, electrons, positrons, kaons and $\gamma$-rays[14,15], which predominantly lose energy via ionization of atoms in the sample and lead to a fast, isochoric, high-energy deposition that penetrates deeply into the meteorite sample. By measuring the surface vibrations of the sample using Laser Doppler Vibrometry (LDV), the response of the iron meteorite sample resulting from the corresponding thermally-induced stress wave was measured in real time for 27 successive

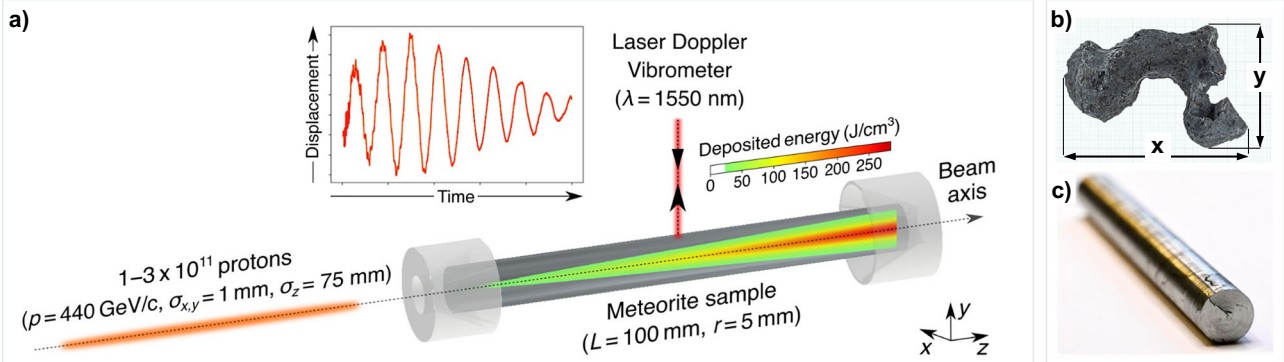

**Fig. 1 | Experimental setup. a** Protons with momenta 440 GeV/c are extracted from CERN's Super Proton Synchrotron in a Gaussian-shaped bunch ($\sigma_{x,y} = 1$ mm, $\sigma_z = 75$ mm) containing $1 - 3 \times 10^{11}$ protons. The cylindrical meteorite sample ($L = 100$ mm, $r = 5$ mm) is held in the beam path using conical supports and irradiated along its central axis. The protons generate hadronic and electromagnetic showers of secondary particles that deposit thermal energy via ionization losses in a quasi-adiabatic fashion. The superimposed colormap shows the expected energy deposition profile obtained from a Monte Carlo simulation (conducted using code FLUKA[33]; the energy-deposition data are provided in Supplementary Data 1 and are also available on Zenodo) when the sample is irradiated with $3 \times 10^{11}$ protons. Radial vibrations and deformation of the sample due to the induced thermal stress are measured in real-time using a laser to perform Laser Doppler vibrometry (LDV) with $\lambda = 1550$ nm wavelength. **b** Campo del Cielo meteorite: surface of raw sample (topview) with visible inclusions dimensions are $x = 21.4$ cm and $y = 14.3$ cm. **c** Meteorite sample cut into a cylindrical shape with dimensions of 1 cm in diameter and 10 cm in length.

beam irradiations (setup shown in Fig. 1). While longitudinal oscillations are coupled to radial ones via the Poisson ratio and are non-negligible, the current diagnostics set-up only captured the radial material response, since it reflects most sensitively changes in the material's yield strength, and the Poisson ratio was not expected to change significantly. In addition, the gathered material response data have been used to derive the fraction of the primary beam kinetic energy converted into bulk kinetic energy of the sample, providing the momentum transfer. As such, the irradiation of the meteorite sample by the proton beam can also be understood as a laboratory-based surrogate used to test the efficiency of particle beam-based asteroid maneuvers - a not-yet-explored method for asteroid deflection.

The maximum primary proton beam intensity is chosen to be $3 \times 10^{11}$ protons at 440 GeV in a single-bunch with a Gaussian transverse profile ($\sigma_{x,y} = 1$ mm) and duration of $\sigma_t = 250$ ps to study the plastic deformation regime, isolated from solid-to-solid phase transitions which are expected at higher energies.

The intensity of the primary proton beam used for each sample irradiation is shown in Fig. 2, and examples of the LDV raw data, showing the radial surface displacement as a function of time, are shown in Fig. 3. For the first 10 shots onto the sample, the primary beam intensity is $1 \times 10^{11}$ protons, and damped oscillatory behavior is observed in the radial displacement of the sample surface. When the primary beam intensity is increased to $3 \times 10^{11}$ protons, non-oscillatory behavior is suddenly observed. This behavior persists even when the beam intensity is reduced to $1 \times 10^{11}$ protons for 3 shots. Eventually, oscillatory behavior returns. Our experiments provide a detailed view of how metal-rich asteroid material responds dynamically to high-energy irradiation. Using the thermal stress model introduced by Bertarelli et al.[16], we estimate that proton beam intensities of $1 \times 10^{11}$ protons induce a peak thermal stress $\sigma_{thermal}$ of ~40 MPa, while intensities of $3 \times 10^{11}$ protons result in stresses of up to 120 MPa (see "Methods, Thermal stress model of the meteorite").

To assess the mechanical response of the meteorite material, we compared these stress levels to reported yield strength values for iron

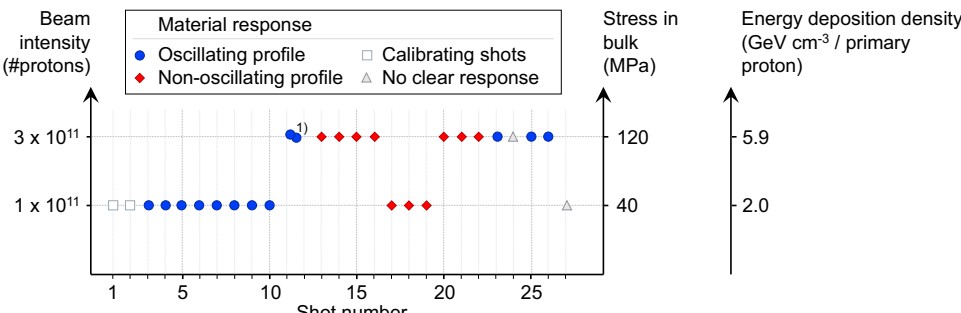

**Fig. 2 | Material response per beam shot.** Material response based on displacement graph profiles per proton beam shot, distinguished by beam intensity (in number of protons), stress in bulk (in MPa), and energy deposition density (in GeV cm$^{-3}$ per primary proton). Each symbol represents one beam shot performed on the meteorite sample. Blue circles indicate shots that produced a clear oscillating profile in the Laser Doppler Vibrometry (LDV) signal. Red diamonds denote non-oscillating profiles. Open gray squares mark calibrating shots, and open gray triangles correspond to shots with no clear response. The beam intensity (number of protons per pulse) is shown on the left axis, with corresponding peak bulk stress (right, middle axis) and energy-deposition density (right axis) derived from Geant4 simulations. The 27 beam shots were distributed over ~7 h, with several minutes between shots. This allowed sufficient thermal and mechanical equilibration of the meteorite sample before each irradiation. The LDV recorded data with an acquisition frequency of 4 MHz for the first 5 ms after the beam shot trigger, covering the time period during which all oscillations fully decayed. (1) Cumulated measurement data of two shots, no separate data available for individual shots 11 and 12. Source data are provided in the Supplementary Information and as a Source Data file.

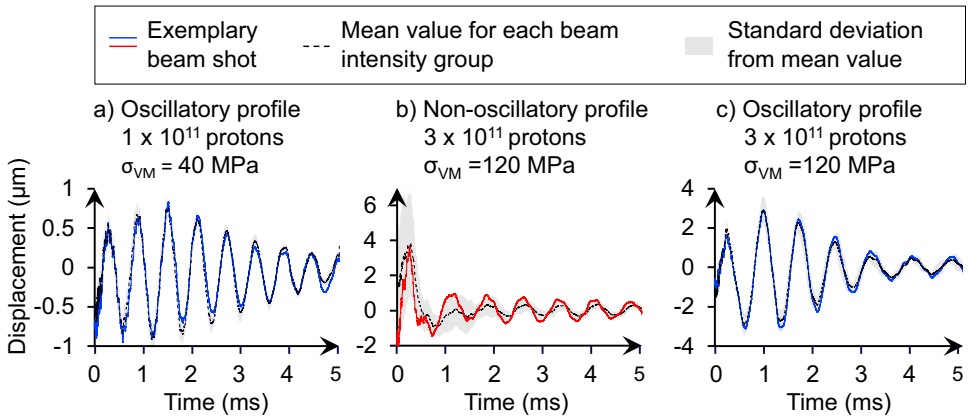

**Fig. 3 | Radial meteorite surface displacement measured by LDV.** Surface displacement in μm over time (ms), captured via Laser Doppler Vibrometry (LDV). Each panel shows the displacement response of the meteorite sample following a single SPS proton-beam impact. Blue lines represent oscillatory response profiles, red lines correspond to non-oscillatory profiles, the black dashed line indicates the mean displacement for all profiles shown in each panel, and the gray shaded area denotes the corresponding standard deviation. **a** Oscillatory response at low beam intensity ($1 \times 10^{11}$ protons, $\sigma_{thermal} = 40$ MPa), shown for example beam shot 4 (blue line), with mean value (black dashed line) and standard deviation (gray area) calculated from beam shots 3–10. **b** Non-oscillatory (plastic) displacement profile at higher beam intensity ($3 \times 10^{11}$ protons, $\sigma_{thermal} = 120$ MPa), shown for example beam shot 14 (red line), with mean and standard deviation from beam shots 13–22. **c** Reappearance of oscillatory behavior at high intensity ($3 \times 10^{11}$ protons), shown for example shot 23 (blue), with mean and standard deviation based on shots 11/12 (cumulative), 23, 25, and 26. Note: vertical scaling differs between graphs for visual clarity.

meteorites. Local yield strength measurements from nanoindentation of Kamacite yield values of ~350 MPa[3], which we denote as $\sigma_{y,local}$. In contrast, yield strength values derived from ram-pressure modeling of meteor breakups in Earth's atmosphere are significantly lower, around 50 MPa[4], reflecting an effective macroscopic response ($\sigma_{y,bulk}$).

If $\sigma_{y,local}$ were directly applicable to our beam experiments, the measured thermal stress would remain well below the yield threshold, and no plastic deformation would be expected. However, our LDV data clearly indicate a transition in oscillatory behavior at higher beam intensities, consistent with the onset of plastic deformation.

This discrepancy reflects the fundamental difference between local and bulk yield strength measurements: nanoindentation captures micromechanical response under quasi-static loading, while bulk estimates account for internal structure, phase boundaries, and inertial coupling across the heterogeneous meteorite volume. To bridge these regimes, we introduce a scaling factor:

$$f = \frac{\sigma_{y,\,local}}{\sigma_{y,\,bulk}} \qquad (1)$$

$\sigma_{y,local}$ is the local yield strength and ($\sigma_{y,bulk}$) is the bulk yield strength. The factor $f$ has a value of ~7 and can be interpreted as a stress amplificiation factor resulting from internal redistribution of von Mises stress under internal inertial degrees of freedom, consistent with the dynamic composite-like behavior[17] of meteorites proposed by Mulford et al.[5].

Applying this factor to the peak thermal stress $\sigma_{thermal}$ implies that the Kamacite phase effectively experienced stresses of $f \cdot \sigma_{thermal}$, i.e., 280 MPa for low and 840 MPa for high beam intensities. These values are consistent with the LDV signal profiles and with the threshold defined by $\sigma_{y,local}$.

At low intensities (280 MPa), the stress remains below the local yield strength, and oscillations persist. At high intensities (840 MPa), the stress exceeds the yield strength, oscillations collapse, indicating transition into the plastic deformation regime.

This transition allows us to calculate the plastic deformation energy $E_{plastic}$, which we assume is fully converted into dislocation generation. Based on the dislocation line energy in Kamacite (see "Methods, Calculation of dislocation density"), we estimate an increase in dislocation density by a factor of ~6, compared to the typical pre-irradiation value of $10^{10}$ m$^{-2}$[18]. Since yield strength scales with the square root of dislocation density, we estimate that $\sigma_{y,local}$ increased to ~875 MPa post-irradiation–a factor of 2.5 increase. This inferred hardening, consistent with the recovery of oscillatory LDV profiles at elevated stress, is hereafter quantified by the hardening factor $h$. This assumption is further supported by the well-known fact that Kamacite reacts strongly with an increase in yield strength at a much lower absolute increase in dislocation density than typical metals[13]. As an additional consistency check, we employed the Grüneisen approach to estimate the local pressure in the region of maximum energy deposition (see "Methods, Calculation of local pressures with Grüneisen parameter"). The resulting pressures of 2–3 GPa (for temperature rises of 300–440 K) are roughly an order of magnitude higher than those predicted by the Bertarelli et al. thermal-stress model, yet remain well below any known equilibrium phase-transition thresholds (e.g., refs. 19,20). This supports the interpretation that the absorbed energy predominantly contributed to plastic deformation.

We then extend this hardening estimate to the bulk response: starting from 50 MPa and applying the same factor $h = 2.5$ suggests a new bulk yield strength of 125 MPa. This again exceeds the 120 MPa thermal stress at high intensity and explains the re-emergence of stable oscillations observed in later LDV measurements.

The predicted displacement amplitudes based on energy deposition profiles systematically underestimate the observed values–unless corrected by the same factor $f$, reinforcing our

hypothesis of internal inertial dynamics and validating the role of structural heterogeneity in amplifying stress locally.

COMSOL Multiphysics simulations of the temperature profile in the meteorite sample, which used the energy deposition profile from Geant4, a Monte Carlo-based particle transport simulation toolkit developed at CERN, as input, were performed[21–23]. The simulations yielded a temperature rise of 4.95–5.9 K at the meteorite surface, closely matching the ~4 K measured by the PT100 sensor located opposite the LDV focal point on the sample surface.

Based on the measured temperature increase, we computed the local thermal strain using the linear thermal expansion relation, $\varepsilon = \alpha \cdot \Delta T$, with a thermal expansion coefficient of $\alpha = 11.1 \times 10^{-6}$ K$^{-1}$. The corresponding thermally induced radial displacement–assuming a purely elastic response and using the sample diameter as the reference length–amounts to 0.44 µm. This value is ~6.5 times smaller than the average displacement measured by the LDV, consistent with the scaling factor $f$ and reinforcing the hypothesis of internal inertial dynamics within the meteorite sample.

Together, these findings yield three central insights:

1. They explain the discrepancy between local and bulk yield strength measurements and quantify a consistent scaling factor;
2. They demonstrate a dynamic hardening path under high strain rate irradiation;
3. They show that mechanical parameters such as yield strength evolve in real time under volumetric energy deposition and should not be treated as static in planetary defense simulations.

This last point is of particular relevance to impact modeling and asteroid deflection strategies, where mechanical parameters are often assumed to be fixed or tabulated.

Moreover, this hardening process is not limited to the stress regimes explored here. Higher beam intensities could drive further increases in yield strength or even induce solid-solid phase transitions, consistent with observations from meteorite recovery and planetary core modeling, where transitions to $\varepsilon$-phase iron occur under extreme conditions[8,24]. This is further confirmed by more recent results, e.g., from Torchio et al.[20]. But these would at least need 12–13 GPa pressure in the stress wave for a transition to the martensitic phase.

Finally, to interpret the frequency spectra observed in the LDV data, we consider the fundamental geometrical scale of the sample, defined by its 10 mm diameter (see "Methods, Meteorite sample and target envelope"). The fundamental radial frequencies are governed by the material's elastic properties and wave propagation speed. However, due to the heterogeneous nature of the meteorite–composed of alternating Kamacite and Taenite phases–the effective speed of sound is not uniform. Reported values range from ~5500 m/s in Kamacite to 3000 m/s in Taenite[3]. As a result, the shortest travel-time paths and local resonance conditions are strongly influenced by the spatial distribution of Taenite inclusions.

Despite this complexity, the observed Fast Fourier Transform (FFT) spectra consistently show dominant frequencies in the 375–450 kHz range (see Fig. 4), which is in good agreement with the expected fundamental radial modes based on the sample geometry and effective sound velocity. This further supports the interpretation that the measured oscillations are governed by volumetric stress-wave propagation and are sensitive to the internal structure of the meteorite sample.

We also observed an unexpected damping phenomenon: the strain amplitude-dependent attenuation of LDV signals. For low-intensity shots, the decay of oscillations is well described by an exponential function. However, high-intensity shots show systematic deviations from exponential decay, indicating the presence of higher-order damping terms. Regression analysis of the residuals (Fig. 5) reveals that these deviations are correlated with oscillation amplitude,

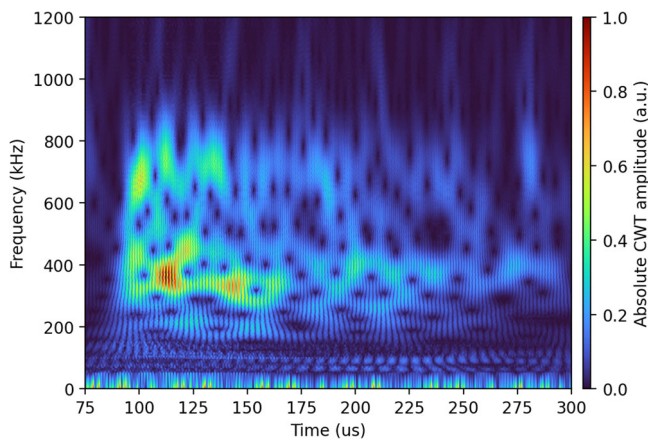

**Fig. 4 | FFT overlay view.** FFT spectrogram of the LDV signale (beam shots 5–10), showing dominant frequency components in the 375–450 kHz range shortly after beam impact. The persistent harmonic structure reflects coherent radial oscillation modes.The corresponding raw data (provided as Source Data, i.e., the TDMS file for each individual shot) are available on Zenodo.

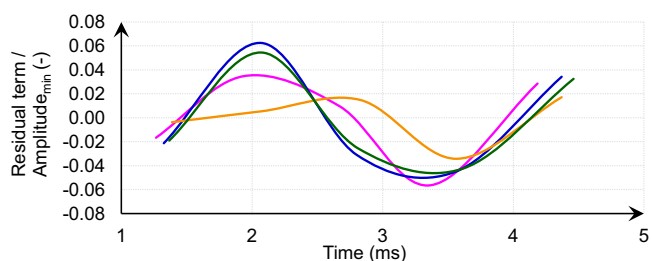

**Fig. 5 | Amplitude-adjusted residual terms per beam shot.** Curves showing amplitude-adjusted residual terms, i.e., residual terms divided by amplitude minima per individual beam shot, respectively over time in ms, for high-intensity beam shots 11/12 (pink), 23 (blue), 25 (orange), and 26 (green) with $3 \times 10^{11}$ protons, resulting in a stress of $\sigma_{thermal} = 120$ MPa. Residual terms are defined as the difference between the measured displacement and the best exponential fit data.

strongly suggesting a strain-dependent damping effect–an observation typically associated with advanced composite materials[17].

This damping behavior might be linked to dislocation dynamics[17] within the hardened Kamacite matrix and is of particular relevance for planetary defense: strain-dependent damping prevents resonant amplification and self-destruction under repeated excitation, making the material inherently more resilient to follow-up perturbations.

Together, these results suggest that high-energy proton irradiation not only hardens iron meteorite material but may transform it into a composite-like structure with improved damping characteristics. This behavior was anticipated in early modeling efforts[5], but we provide a real-time experimental confirmation.

Moreover, this implies that much larger amounts of energy can be deposited into asteroid material than previously assumed–without structural failure. This opens yet unexplored possibilities for nuclear energy-density asteroid deflection techniques, where deep energy coupling is desired without fragmentation. Future work will explore this scenario in detail, including implications for momentum transfer efficiency, phase transition thresholds, and the scalability of the hardening process.

## Methods

### Meteorite sample and target envelope
A piece of the Campo del Cielo iron meteorite was cut into a cylindrical shape of 10 cm in length and 1 cm in diameter, having a mass of 60 g.

Campo del Cielo is known to be composed mainly of iron (92.7%) and nickel (6.15%), as well as cobalt (0.42%), carbon (0.37%) and phosphorous (0.28%)[25] with silicon, titanium, vanadium, gallium, copper, and sulfur present at trace levels[26]. The two mineral phases are Kamacite - having a body-centered cubic (BCC) lattice -, and Taenite - having a face-centered cubic lattice.

The proton beam deposition behavior is principally determined by the density of the irradiated material. With 7.86 g/cm³ for iron and 8.90 g/cm³ for nickel, the weighted average density utilized in the Monte Carlo simulations was 8.06 g/cm³. The minor density variations within the meteorite sample from the ppm-level inclusions like e.g., phosphorous are not expected to play a significant role in the elastic wave behavior of the material.

The meteorite sample was irradiated by the proton beam along its longitudinal axis. Both ends of the target envelope were protected by 2 mm thick SIGRADUR® glassy carbon windows. The target envelope was made of aluminum with a wall thickness of 15 mm.

### Pulse intensity and shot number selection
The experimental design was guided by the objective to probe and compare distinct mechanical response regimes of iron meteorite material under high-energy proton irradiation. Two pulse intensities were selected to this end:

- A lower intensity of $1 \times 10^{11}$ protons per pulse was chosen to keep the sample response within the elastic regime, ensuring minimal permanent deformation.
- A higher intensity of $3 \times 10^{11}$ protons per pulse was employed to deliberately trigger the onset of plastic deformation, enabling a direct comparison between elastic and plastic material behavior.

As the experimental setup constitutes a unique configuration that has not previously been implemented for meteorite irradiation, the higher intensity level was conservatively chosen. It remained well below thresholds associated with phase transitions or structural failure, ensuring safe operation within established facility limits. The pulse list and the corresponding detailed beam parameters are available in the Supplementary Information provided with this article.

The number of shots per condition was determined to strike a balance between statistical significance and safety. The total integrated beam energy was carefully kept below levels that could cause irreversible damage to the sample or compromise experimental infrastructure. This allowed for robust characterization of the material's response in both regimes without exceeding risk thresholds.

### Laser Doppler Vibrometry (LDV)
The LDV used, consisted of an acquisition unit connected to a manual-focusing head without electronics, allowing for a focal distance of 15 mm to 5 m. The acquisition unit contains the necessary electronics to convert the analog signals from the LDV head into digital measurements of displacement, velocity, and acceleration. A PXI (PCI eXtensions for Instrumentation) acquisition system was also connected to the LDV acquisition unit to trigger the recording and make the LDV data available remotely. In this work, a PXIe-6124 (National Instruments) digitizer card with 16-bit resolution and selectable input ranges ( ±1 V, ±2 V, ±5 V, ±10 V) was employed to record the analog output signals of the LDV. The LDV acquisition unit and the LDV head were connected by a pair of single-mode polarization-maintaining fibers that ran through the wall of the tunnel, ensuring that the electronics were placed away from radiation. In the experimental set-up, the distance between the LDV head and the focal spot on the meteorite was 550 mm, with a focal spot diameter on the order of 100 μm. The data quality and reliability of the LDV laser on the meteorite surface was tested in several iterations before

the experiment. The tests confirmed that the initially curved surface of the meteorite sample, with its small radius of 5 mm, made the alignment of the LDV Laser very difficult and prone to errors. To ensure high-quality LDV signal acquisition, the convex curvature of the cylindrical meteorite surface was flattened by carefully removing ~1 mm of material with a fine-grained steel file, creating a polished, flat surface measuring about 4 mm. This preparation enabled 100% signal strength in the LDV measurements.

The present study focused exclusively on radial surface displacement, which directly reflects changes in yield strength. While longitudinal and surface wave modes also exist, they were not captured due to experimental access limitations. However, the return to clean oscillatory response after strain hardening suggests that the dominant dynamics were captured adequately by the radial LDV measurements. All utilized LDV displacement data are available as raw data under Zenodo.

## Temperature measurement with PT100

The temperature of the meteorite surface was measured with a 4-wire PT100 platinum resistance thin film detector, attached to the meteorite sample on the opposite side of the LDV focal spot. Temperature data were gathered continuously at a frequency of 1 Hz. Temperature data are available as raw data, accessible on Zenodo.

## Monte Carlo simulations of energy deposition profile

Simulations of the temperature profile of the meteorite cylinder were performed using COMSOL Heat Transfer in Solids module[21]. Energy deposition data from GEANT4 simulations was used as input for the heating source[22,23]; the corresponding source data are provided in Supplementary Data 3 of this paper. The meteorite was given an initial uniform temperature of 293.15 K. The temperature increase closest to the position of the PT100 temperature was found to be between 4.95 and 5.9 K. (see Fig. 6).

## Thermal stress model of the meteorite

The applied thermal stress was $\sigma_{thermal}$ estimated from the thermal expansion relation as e.g., presented in ref. 16

$$\sigma_{thermal} = E \cdot \alpha \cdot \Delta T \tag{2}$$

where $E$ denotes the Young's modulus, $\alpha$ the thermal expansion coefficient and $\Delta T$ the temperature increase. To assess material failure and plastic deformation onset, the resulting thermal stress was expressed as the von Mises equivalent stress. For the calculation, literature values for Kamacite and Taenite were utilized, as listed in Table 1.

Our approach focuses on comparing these analytically calculated stress levels to the yield strength values inferred from the LDV-measured surface vibrations. The observed consistency between the analytically predicted stress thresholds and the experimentally detected transitions—from elastic behavior to plastic deformation, and subsequently to oscillatory behavior indicative of strain hardening—is fully consistent with the selected material parameters.

## Calculation of local pressures with Grüneisen parameter

To estimate the transient pressure rise associated with ultrafast, quasi-isochoric heating during the proton pulse, we applied the Grüneisen parameter. The Grüneisen parameter $\gamma$ relates the thermal pressure $P_{th}$ to the energy density $E_{th}$ through

$$P_{th} = \gamma E_{th} = \gamma \rho c_v \Delta T, \tag{3}$$

where $\rho$ is the mass density, $c_v$ the specific heat at constant volume, and $\Delta T$ the temperature increase derived from the COMSOL simulations[21].

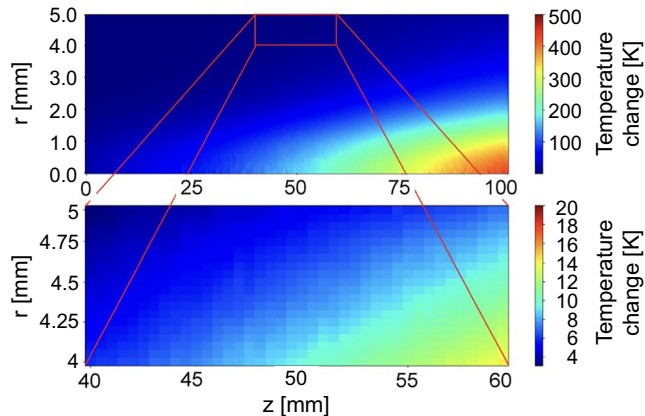

**Fig. 6 | Temperature profile in the meteorite.** Top: Radial temperature profile showing the temperature at $t = 1$ ns. Bottom: Radial temperature profile in 1 mm × 20 mm near the location of the PT100 temperature sensor used in the experiment.

### Table 1 | Material properties of Kamacite and Taenite

| Property | Value | Unit | Reference |
|---|---|---|---|
| Young's modulus (Kamacite), $E_{Kamacite}$ | 241 | GPa | 28 |
| Young's modulus (Taenite), $E_{Taenite}$ | 68 | GPa | 28 |
| Yield strength (Kamacite), $R_{m,Kamacite}$ | 350 | MPa | 3 |
| Yield strength (Taenite), $R_{m,Taenite}$ | 935 | MPa | 3 |
| Specific heat (20–100 °C), $c_p$ | 0.461 | J/g · K | 29,30 |
| Thermal expansion (20–100 °C), $\alpha$ | $11.1 \times 10^{-6}$ | K⁻¹ | 31 |
| Poisson's ratio, $v$ | 0.3 | – | 32 |

The Grüneisen parameter was calculated for the dominant kamacite phase using

$$\gamma = \frac{\alpha_V K_T}{\rho c_v}, \tag{4}$$

where $\alpha_V$ is the volumetric thermal expansivity and $K_T$ the isothermal bulk modulus. Using the values listed in Table 1 with $\alpha_V = 3 \times \alpha = 3 \times 11.1 \times 10^{-6}$ K⁻¹, $E = 240$ GPa, $v = 0.3$, $\rho = 8060$ kg m⁻³, and $c_p = 461$ J kg⁻¹ K⁻¹ (approximating $c_v \approx c_p$), we obtained the bulk modulus $K_T = E/[3(1 - 2v)] = 200$ GPa and Grüneisen parameter $\gamma$ of -1.8.

Assuming local heating of $\Delta T = 300-440$ K within the beam-intercepted volume, the corresponding local pressures are

$$P_{th} = \gamma \rho c_v \Delta T. \tag{5}$$

The resulting thermal pressure is ~2–3 GPa. This range exceeds by roughly one order of magnitude the stress values predicted by the analytical thermal-stress model of Bertarelli et al.[16], yet remains well below any known equilibrium phase-transition thresholds for Fe–Ni alloys[20]. For comparison, Torchio et al.[20] report that in Fe–Ni alloys the bcc → hcp ($\alpha \to \varepsilon$) transition begins at pressures above ~12 GPa and completes by ~17 GPa for compositions of about 20 wt% Ni, while shifting to even higher pressures (>100 GPa) with increasing Ni content. The average Ni concentration in our iron meteorite sample is significantly lower (around 6.15 wt% overall, around 5.5 wt% for the dominant bcc kamacite phase). Consequently, the 2–3 GPa local pressures estimated here lie well below any known equilibrium phase-transition thresholds, and the deposited energy is therefore

interpreted as contributing primarily to plastic deformation and defect production, rather than to melting or structural phase change.

It should be noted that the LDV measurements provide volumetrically averaged parameters, as discussed in the LDV subsection of the "Methods" section. The localized zone of maximal temperature increase, defined by $\Delta T \geq 300\,K$ (as assumed for the local heating), accounts for only ~0.7% of the total cylinder volume, whereas the resulting thermal-stress wave propagates throughout the entire sample. Consequently, the experimentally observed LDV response represents a spatially averaged response of these local processes.

### Calculation of plastic deformation energy

The LDV measurements were taken at a single point on the surface of the sample, halfway along its cylinder axis. Since this location does not capture the spatial variation in energy deposition across the bulk, we applied a correction based on temperature profiles obtained from simulations using COMSOL Heat Transfer in Solids module. The LDV-derived energy values were scaled by the ratio between the simulated local temperature and the temperature measured by a PT100 sensor[21]. This procedure allowed us to estimate the total energy that contributed to plastic deformation across the entire sample volume. The final result of this step was a plastic deformation energy of $E_{plastic} = 360\,\mu m$, attributed to the generation of dislocations in the meteorite sample.

### Calculation of dislocation density

The estimation of the dislocation density increase proceeded in two steps: (1) determination of the energy converted into plastic deformation, and (2) calculation of the corresponding dislocation density based on dislocation line energy.

Step 1: To estimate the increase in dislocation density, we first determined the amount of energy converted into plastic deformation. We assumed that any energy not stored in oscillatory modes was dissipated through plastic mechanisms. Accordingly, the plastic deformation energy was calculated as $E_{plastic} = E_{oscillating} - E_{non-oscillating}$.

To evaluate these energy terms, we extracted the average kinetic energy from the displacement profiles. For non-oscillating diagrams, we applied the classical kinetic energy expression $E = \frac{1}{2}mv^2$, where the velocity $v$ was estimated from the small residual displacements. For oscillating diagrams, the energy stored in the oscillatory motion was calculated as $E = m\pi^2(vA)^2$, where $m = 60\,g$ is the mass of the sample, $v = 1.5\,kHz$ is the average oscillation frequency, and $A$ is the measured amplitude. For low-intensity shots, the average amplitude was $A = 0.85$ m; for high-intensity shots, $A = 2.9\,\mu m$. Using these values, the plastic deformation energy was calculated, amounting to $E_{plastic} = 360\,\mu J$.

Step 2: We then calculated the dislocation line energy $E_{disloc}$, i.e., the energy required to generate a unit length of dislocation in Kamacite. This is given by

$$E_{disloc} = \frac{1}{2}Gb^2, \qquad (6)$$

where $G$ is the shear modulus and $b$ is the magnitude of the Burgers vector[27]. Kamacite has a BCC lattice structure with a lattice constant $a$ of ~2.88 Å, resulting in $b = \frac{\sqrt{3}}{2}a$ with a value of ~1.44 Å. Using $G = 75\,GPa$, we find a value for $E_{disloc}$ of about $7.78 \times 10^{-10}$ J/m.

Assuming that the entire plastic deformation energy is stored in newly formed dislocation lines, the total dislocation length created is given by $L_{disloc,total} = E_{plastic}/E_{disloc}$, yielding a value of ~$4.63 \times 10^5$ m.

Dividing this by the sample volume yields an estimated increase in dislocation density of ~$6.1 \times 10^{10}$ m$^{-2}$, consistent with values reported for moderate hardening in BCC Kamacite.

### Time-frequency analysis via continuous wavelet transform

To resolve the temporal evolution of vibrational modes following proton beam impact, we performed a time-frequency analysis of the radial displacement signal obtained from LDV. A continuous wavelet transform (CWT) was applied using a complex Morlet wavelet. The analysis was applied to a 225 μs time segment (75–300 μs after beam impact), capturing the key dynamic response of the meteorite. Frequencies between 50 kHz and 1.5 MHz were evaluated in 5 kHz steps by translating each frequency into the corresponding wavelet scale, based on the sampling rate and wavelet center frequency. Before applying the CWT, the radial displacement signal was bandpass-filtered between 5 kHz and 1.5 MHz to suppress baseline drift and instrumental noise, ensuring that only physically meaningful frequencies were retained for analysis.

### Analysis of Campo del Cielo sample

The average composition and material characteristics were optically analyzed using scanning electron microscopy (SEM) and energy-dispersive X-ray spectroscopy (EDS). Pictures of the optical analysis are included in the Supplementary Material.

## Data availability

Source data are provided with this paper. Additional raw data supporting the findings of this study are available in the public repository at https://doi.org/10.5281/zenodo.17582347.

## Code availability

Simulations were performed using FLUKA, COMSOL Multiphysics (v6.2; https://www.comsol.com), and the open-source Geant4 Monte Carlo toolkit (https://geant4.web.cern.ch) developed at CERN. A custom analysis script was used to process and visualize the Laser Doppler Vibrometry (LDV) voltage output data over time. The script is available from the corresponding author upon request.

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

## Acknowledgements
We thank Thomas Zurbuchen for early and continued discussions on the fundamental concept underlying the experiment. We thank Nancy Chabot, Lindey Elkins-Tanton, Ludovic Ferrière and Stefan Hölzl for the early exchange on choosing adequate meteorite properties as well as Pierre Antonin for providing a piece of Campo del Cielo. The raw meteorite was cut using the machine shop at the university of Münster, kindly coordinated and supported by Bastian Gundlach and Markus Patzek. Erika Griesshaber and Wolfgang Schmahl kindly gave us access to the Electron Microscopy Laboratory at the Department of Earth- and Environmental Sciences, Applied Crystallography and Materials Science of the Ludwig-Maximilians-University of Munich. The team of the Department for Civil Engineering and Environmental Sciences, University of the Bundeswehr Munich with Wolfgang Saur, Christian Thienel and Alexandra Widuch also gave us access to their laboratory, allowing for initial SEM and computerized tomogrpahy (CT) images of the meteorite sample. We thank the team from CAE Simulation & Solutions for designing and engineering the target envelope. We thank John Bekx for carefully reading the manuscript. This project has received funding from the European Union's Horizon Europe research and innovation program under grant agreement No 101057511. The work of G.G. and E.E.L. was supported by UKRI under grant no. EP/Y035038/1.

## Author contributions
M.B. and K.-G.S. conceptualized the study. G.G. proposed to incorporate the study into the FB II experimental campaign. M.B. and K.-G.S. wrote the manuscript. N.C., P.S., and E.S. contributed to writing and revising the paper. C.D.A. supported the work from initial simulations to manuscript completion (including visualization of the experimental setup), with a consistent focus on the key messages. G.G. proposed incorporating the meteorite study into the Fireball II collaboration's experimental campaign and contributed to sharpening the key messages and improving the manuscript structure. M.B. and K.-G.S. carried out the data analysis with support from G.G. and P.S. M.B. and K.-G.S. designed the experiment with support from C.D.A., R.B., N.C., A.M.G., G.G., J.W.D.H., P.S., E.S. M.B. and K.-G.S. carried out the experiment with support from A.M.G., M.A.P., P.S., and E.S. E.S. and M.A.P. selected and provided the experimental diagnostics with support from M.A., E.K., and M.L. A.M.G., E.S., J.-M.Q., and A.E.R. installed and tested the diagnostics and the meteorite envelope. J.-M.Q. treated and shaped the meteorite piece. P.A., E.M.A., C.D.A., B.L., N.C, P.R., and P.S. performed numerical simulations. P.A., P.J.B., F.D.C., B.T.H., E.E.L., B.R., S.S., L.O.S., V.S., and S.Z. contributed through experimental integration meetings that were critical for planning, coordinating, and implementing the experimental setup.

## Competing interests
The authors declare no competing interests.
