## [Transparent Peer Review file · Nature Communications]

Dynamical development of strength and stability of asteroid material under 440 GeV proton beam irradiation

Corresponding Author: Ms Melanie Bochmann

Version 0:

Reviewer comments:

Reviewer #1

(Remarks to the Author)

In this paper, the authors study the dynamic strength on iron meteorites under irradiation via a proton beam at CERN. Using Laser Doppler Vibrometry techniques, the authors are able to measure the stress wave in the sample in time over successive irradiation events. They find that under these proton irradiations, the yield strength of the material significantly increases, potentially up to seven times the standard yield strength of Kamacite. While these results are intriguing, it is unclear how this extends to planetary defense in general. Further, the analysis performed on the LDV data is slightly ambiguous and more details on the experimental set-up and analysis are needed prior to publication.

Major Comments

-Some major details about the irradiation timing are needed in the paper. While the authors list that the pulse duration of the proton beam is 250ps, they do not list the frequency at which the pulses are incoming. This is relevant to the LDV analysis – all the LDV data are shown out to 5 ms but is the sample allowed to fully stop its motion or are the pulses arriving faster than 5 ms?

-In the methods, the authors state that a notch was filed into the sample to increase the LDV signal – how does this notch effect the LDV? Surface roughness could change the signal response from LDV resulting in difficult to analyze data – was the sample polished to remove any roughness? If not, how could this roughness affect the results?

-In reference to Figure 4: on line 232-233, it is said that “show in average larger deviations for the high-intensity shots than for low-intensity shots” with a reference to the methods for more details. Looking at Figure 4, it appears to me that the low-intensity shots show larger deviations than the high-intensity. Is this simply a typo or is there something more fundamental that I am missing.

-Additionally, why does the sample return to oscillating as it did initially, even during the higher intensity pulse? Does this indicate that the hardening is complete or is there something else going on? Later in the discussion, it seems to imply that the material was not fully hardened so some more details would be useful.

-The authors present two potential strength hardening scenarios – one in which the sample increases in yield strength by a factor of ~ 2 and one by a factor of ~7. As noted in the paper, for planetary defense simulations, the choice of yield strength is of great importance. It would be valuable to explain which one of these two scenarios are the best for simulations as the authors suggest that proton beam irradiation could be a viable planetary defense mitigation method.

-Some discussion would be useful on how these results extend to other planetary defense methods. As of now, proton beam irradiation is not one of the common techniques for asteroid deflection. Does this radiation induced hardening extend to other forms of radiation – such as that produced from a nuclear explosive device – or is it unique to GeV protons? Further, is this unique to iron-nickel meteorites or would you expect to see this hardening in more common rocky asteroid materials?

Minor Comments

-Lines 49 – 54 are a single sentence which is quite long and somewhat difficult to parse. I would suggest separating it into two, which could add further emphasis on the Planetary Defense aspects you are highlighting in the paper.

-Line 98 – no ‘-’ in inhomogeneous

-Figure 4 – placing both data sets on the same y-axis would help with comparison.

-Line 324 – weighting is misspelled.

Reviewer #2

(Remarks to the Author)

Please consider the attached pdf file for comments.

Reviewer #3

(Remarks to the Author)

Overview:

The authors examine the effect of high-energy (440 GeV) proton irradiation on a meteorite sample using the HiRadMat facility at CERN. The subject is of broad scientific interest, and the experiment itself is quite interesting. The authors observed a change in the structural dynamic response of the cylindrical meteorite sample, as measured with a laser doppler vibrometer, most of the time occurring when the beam irradiance was increased. The authors attribute this change in signal to energy loss and assert it arises from plastic deformation. Based on this interpretation, they estimate the amount of strain hardening the sample may have experienced. While the authors reasonably show that this is plausible, it is not convincing without further discussion to eliminate other possibilities and/or without further quantification of uncertainties.

Major:

The single LDV measurement directed orthogonally to the rod should be primarily sensitive to flexure and/or radial dilation, whereas longitudinal modes should also exist (see Ref.22) yet are not accounted for in the analysis. Rather, all energy is assumed to be contained within the radially directed wave, which may not be strictly correct. In particular, there is no comparison of the wave energy to the beam energy for the so-called “elastic case”, or to some other control, to prove this methodology or to otherwise quantify uncertainties in determining the quantity of energy dissipated through plastic deformation.

Similarly, there is no discussion of other modes, including the possibility of surface waves, and if those could produce the change in signal pattern at the higher beam irradiance.

The observed resonance frequencies (Fig.3b) are not sufficiently explained. For example, if one assumes a nominal wave speed of ~3000 m/s, the primary resonances (Fig.3a) have wavelength ~1.8 m, which does not appear to correspond to a primary geometric feature of the sample. One wonders if this could indicate leakage of mechanical energy to the fixture (“envelope”).

The initial temperature of the rod before each pulse should be described. Was it allowed to equilibrate? Did the rod always start at room temperature, or did it get hotter with each radiation pulse? What is the effect of successive heating? It may also help to describe the time between radiation pulses.

One would like to see a quantitative measure of dislocation density before and after irradiation, or some other independent measurement, to confirm the author’s hypothesis regarding structural change.

In general, I do not find the explanation of strain hardening convincing. I agree that this is an intriguing possibility with important implications and is worth exploring. However, reaching this conclusion depends on correctly interpreting a very complex series of behavior that occurred sequentially, whereas the experiments only measured one aspect of this behavior or the interpretation may not be correct (see my other comments).

Minor:

The motivation for placing a phase boundary beneath the focal spot of the LDV needs more explanation. In particular, how would a “preferred direction” of the stress wave be determined from this single point measurement? How was it determined that there was no “preferred direction”?

Fig.1b: Scale bar does not exist or is too small to see, which makes interpretation of the image difficult.

Fig.2: There is no discussion of the shots with “no clear response”. Does this imply difficulties with the measurement? What is the implication for accuracy of the other shots?

Fig.2: Authors should check the units in the energy scale; they do not appear consistent with the energy deposition values in Fig.1a.

Line 119—122: This sentence appears overstated, as the present study does not show how proton beams could be used to deflect or maneuver asteroids. The present study only addresses the wave behavior.

Fig.3a: The two right-most of the three graphical panels have the same labels. Consider differentiating them. Also, it is unclear why the response returns to “elastic” on shot 23 even though the beam energy remains high.

Line 178: “peak-induced stress” is not clearly defined.

Line 189: “internal dynamics” is not clearly defined.

Lines 166-167, 178-180, 196-205: Consistency with ram conditions or shock impact conditions is not necessarily expected, as yield strengths are known to be sensitive strain-rate and/or shear conditions.

Fig.5: The meaning of the color bar is unclear, specifically the normalization “per bin of 1 cm³” is sometimes used for energy

but not usually for temperature. Should the units instead be "K/pulse? Or this normalization should be described.

Line 301-303: It does not seem correct to say "the PT100 sensor measured roughly the same amount of temperature increase as for the oscillating diagrams" because the diagrams do not measure temperature. Or the method behind this inference should be explained more fully.

Line 324-325: Why was this weighting performed? Does it imply a systematic error in the model?

Line 329-330: The values (with units) when used in (2) do not appear to give units of energy; they give units of force. Also, it is unclear how the dislocation density was calculated from (2).

Version 1:

Reviewer comments:

Reviewer #1

(Remarks to the Author)

The edits and thoughtful responses provided by the authors have addressed all of my comments/concerns about the paper from the first revision. There are some very minor issues that remain that I have detailed below:

1. Citation 1 (Miles et al) is probably not the best citation for general Planetary Defense, including references to Z-machine experiments. I would suggest either citing the Dearborn and Miller for a general overview of Planetary Defense and early work by Remo on Z-machine experiments.

2. Line 176 has a colon like it is going to provide a definition - I think it should be a period.

Reviewer #2

(Remarks to the Author)

All the comments provided by this reviewer have been carefully addressed and fully incorporated, leading to a substantial improvement in the overall quality of the manuscript. Therefore, the article is now deemed suitable for publication.

Reviewer #3

(Remarks to the Author)

The authors have substantially improved both the rigor and clarity of this work. Several issues still remain as outlined below.

Major comments

1. The temperature map (Fig.6), based on revised, higher-resolution simulations, shows substantially higher peak temperatures than before. Because the heating is nearly instantaneous (250 ps per pulse), one can use the Grüneisen parameter to estimate local pressure of several GPa in the region of maximum energy deposition, which is an order-of-magnitude higher than the peak stress estimated by the authors using the method of Bertarelli based on the response of the entire rod. This reintroduces the question of localized, solid-solid phase transition contributing to the energy dissipation. The authors cite two papers on the high-pressure phase behavior of iron (Jain, Anderson); however, there has been more recent work suggesting Fe and Fe/Ni alloys may undergo fcc to hcp transition under conditions of a few hundred Kelvin temperature rise and a few GPa pressure (as an example only, see Torchio 2020, DOI:10.1029/2020GL088169). Given present uncertainties in phase behavior in this regime, more discussion or analysis on this point seems necessary to distinguish between solid-solid phase transition and the proposed defect production.

2. Following the above, the authors should consider if the response measured with LDV should be contextualized as a volumetrically-averaged result or similar, and the potential influence (or not) of localized heating and stress may need to be discussed.

Minor comments

1. Fig.1b is still missing a scale bar for the inset at upper right, which is necessary for interpreting the image.

2. Abstract, first sentence: "dynamic response" is vague, which makes the statement "an effect that has not been captured until now" not particularly meaningful.

3. Fig.6, fonts are very small and difficult to read on the color bar and elsewhere.

4. Line 293: As a point of style only, "worldwide" seems overly emphatic and unnecessary.

Version 2:

Reviewer comments:

Reviewer #3

(Remarks to the Author)

The authors have addressed all of my concerns, and I recommend publication of this manuscript. I would like to point out, however, that the thermal expansion coefficient stated on line 213 is inconsistent with the value stated in the Methods, Table I. I assume this is a typographical error.

25 September 2025

Dear Reviewers,

We would like to thank you once again for your valuable comments and constructive suggestions on our manuscript entitled “*Dynamical development of strength and stability of asteroid material under 440 GeV proton beam irradiation*” (Manuscript ID: NCOMMS-24-76790).

Following the editor’s and reviewers’ guidance, we have undertaken substantial additional work beyond textual revision. In particular, we have:

Expanded Motivation and Scientific Context:

- We strengthened the motivation by situating our work more clearly in the context of planetary defense and explicitly highlighting the lack of real-time, non-destructive dynamic material response data for asteroid materials.

New result added based on additional Simulations

- In response to the referees’ request for further quantification, we have conducted additional temperature profile simulations in COMSOL that demonstrate the consistency of vibrational and temperature measurements as a new conclusion in the paper (see Discussion).

Clarified Experimental Setup and Methodology:

- We added details on beam parameters (pulse list), sample mounting, and meteorite preparation. We also discussed how the heterogeneous material structure (e.g., Kamacite-Taenite phases) affects wave propagation.

Restructured Results, Discussion and Methods Sections:

To improve clarity, we reorganized the results, discussion, and methods sections:

- The derivation of dynamic hardening is now presented in clearly delineated steps, with supporting references provided.
- We introduced a scaling factor f to quantify the effect of internal inertial dynamics, bridging local and bulk yield strength.
- Calculation assumptions and values are now explicitly listed in the methods section, distinguishing between tabulated data and measured values.
- Terminology for LDV displacement analysis was refined for consistency and clarity.

We believe that these revisions have significantly sharpened the manuscript’s central statements regarding the dynamic development of strength, while also clarifying the overall argumentation. We are therefore pleased to resubmit the manuscript for your consideration.

To facilitate the review process, we have prepared a point-by-point response to each referee’s comments, indicating where revisions were made in the manuscript. We are also providing a comparison document that highlights all changes between the original and revised versions.

We are very grateful for the referees’ detailed feedback and for the opportunity to revise and resubmit our work.

Sincerely,

Melanie Bochmann (on the behalf of all co-authors)

Reviewer #1 (Remarks to the Author):

In this paper, the authors study the dynamic strength on iron meteorites under irradiation via a proton beam at CERN. Using Laser Doppler Vibrometry techniques, the authors are able to measure the stress wave in the sample in time over successive irradiation events. They find that under these proton irradiations, the yield strength of the material significantly increases, potentially up to seven times the standard yield strength of Kamacite. While these results are intriguing, it is unclear how this extends to planetary defense in general. Further, the analysis performed on the LDV data is slightly ambiguous and more details on the experimental set-up and analysis are needed prior to publication.

MAJOR COMMENTS

- 1) *Some major details about the irradiation timing are needed in the paper. While the authors list that the pulse duration of the proton beam is 250ps, they do not list the frequency at which the pulses are incoming. This is relevant to the LDV analysis – all the LDV data are shown out to 5 ms but is the sample allowed to fully stop its motion or are the pulses arriving faster than 5 ms?*

Reply: This question is a very good hint to further clarify the overall frame conditions, in which the beam shots were conducted. The 27 beam shots were distributed over approximately 7 hours, with time intervals of several minutes between shots. This allowed sufficient thermal and mechanical equilibration before each shot. The LDV acquisition window was limited to 5 ms after each pulse, as all oscillations fully decayed within this period. Thus, no overlap occurred between shots.

The detailed pulse list with times was provided in the supplementary material in the previous manuscript version. We added more details on the pulse frequency and irradiation timing in the “results” section, comprised in the caption of figure 2.

- 2) *In the methods, the authors state that a notch was filed into the sample to increase the LDV signal – how does this notch effect the LDV? Surface roughness could change the signal response from LDV resulting in difficult to analyze data – was the sample polished to remove any roughness? If not, how could this roughness affect the results?*

Reply: The sample surface beneath the LDV laser was polished after removing the convex meteorite material. Following this preparation, LDV tests showed 100% signal strength. The reviewer rightly points out that surface roughness plays a crucial role in achieving high-quality LDV signals. We took this into account by adding a remark in the methods section under paragraph Laser Doppler Vibrometry, explaining that a fine-grained steel file was used in order to achieve a polished, plane surface.

Regarding the impact of the notch on the LDV measurement data, we had similar concerns due to the highly inhomogeneous structure of the meteorite material. In both cases—altering the cylindrical shape of the probe and accounting for the material’s internal structure—we would have expected strong high-frequency components in the Fourier-transformed spectrum if such effects were significant. However, no such indications were observed in the data, suggesting that the influence is negligible.

- 3) *In reference to Figure 4: on line 232-233, it is said that “show in average larger deviations for the high-intensity shots than for low-intensity shots” with a reference to the methods for more details. Looking at Figure 4, it appears to me that the low-intensity shots show larger deviations than the high-intensity. Is this simply a typo or is there something more fundamental that I am missing.*

Reply: The remark that this phrasing has potential for misinterpretation is an important clarification and highly appreciated. Figure 4 displays amplitude-adjusted residuals, defined as the residuals divided by the minimum amplitude of each respective beam shot. In contrast, the comparison discussed in the text refers to the unadjusted residuals, i.e., the difference between the measured displacement and the best-fit exponential curve, without any normalization.

To eliminate this ambiguity, the revised manuscript now focuses on the key finding and presents only amplitude-adjusted residuals for high-intensity shots. We have also revised the figure caption accordingly to clarify this distinction.

- 4) *Additionally, why does the sample return to oscillating as it did initially, even during the higher intensity pulse? Does this indicate that the hardening is complete or is there something else going on? Later in the discussion, it seems to imply that the material was not fully hardened so some more details would be useful.*

Reply: We sincerely thank the reviewer for this insightful question, which allowed us to clarify the underlying hardening mechanism in more detail. The return of oscillatory behavior at high beam intensity — following a regime of non-oscillatory displacement — indicates that the material experienced a strain-induced hardening process, rather than reaching a hardening saturation or failure threshold. As shown in Figure 2 and Figure 3, the collapse of oscillations at beam intensities of 3×10^{11} protons corresponds to the onset of plastic deformation, while the subsequent recovery of oscillations reflects a dynamic increase in yield strength resulting from dislocation buildup in the Kamacite phase.

In other words, the sample initially softens due to plastic deformation, but then hardens progressively as dislocations accumulate. This increases the local and bulk yield strength such that the same high thermal stress (up to ~ 120 MPa) no longer exceeds the (now higher) yield threshold, allowing the sample to sustain elastic oscillations again.

This interpretation is supported by the following:

- Dislocation density increased by a factor of ~ 6 , as estimated from energy deposition and dislocation line energy.
- Yield strength scales with the square root of dislocation density, implying an increase of local yield strength from ~ 350 MPa to ~ 875 MPa.
- Applying the same factor to the bulk yield strength yields ~ 125 MPa, which is just above the peak thermal stress — consistent with the observed reappearance of oscillations.
- The LDV spectra before and after hardening show no sign of damage or anisotropic wave propagation, suggesting structural integrity and uniform response.
- GEANT4 simulations also require a correction factor consistent with this hardening behavior to match observed displacements.

Thus, the return of oscillations does not imply that the material is “fully” hardened in an absolute sense, but rather that the current yield strength has risen above the applied stress, stabilizing the elastic response. Additional hardening could still occur under even higher stress regimes, as discussed in the manuscript.

To address this more clearly, we reorganized the manuscript to provide a more structured explanation of the plastic regime, the onset of hardening, and the interplay between the different effects involved. We have now clarified in the “Discussion” section that the observed recovery of oscillations reflects a strain-hardening effect reaching a new dynamic threshold — not an indication of complete hardening or saturation.

- 5) *The authors present two potential strength hardening scenarios – one in which the sample increases in yield strength by a factor of ~ 2 and one by a factor of ~ 7 . As noted in the paper, for planetary defense simulations, the choice of yield strength is of great importance. It would be valuable to explain which one of these two scenarios are the best for simulations as the authors suggest that proton beam irradiation could be a viable planetary defense mitigation method.*

Reply: Thank you for this important remark. It helped us reorganize the Discussion section to more clearly distinguish between the different effects influencing yield strength.

In the revised manuscript, we now emphasize the distinction between two types of yield strength values commonly used in planetary defense modeling:

- Local yield strength ($\sigma_{\text{local}} \sim 350$ MPa): This higher value is based on nanoindentation measurements and reflects the micromechanical response of the Kamacite phase in iron meteorites (Ueki et al.)
- Bulk yield strength ($\sigma_{\text{bulk}} \sim 50$ MPa): This lower value, smaller by a scaling factor $f \approx 7$, is derived from atmospheric ram pressure modeling of meteorite breakup events and accounts for internal structure, phase boundaries, and inertial coupling across the heterogeneous meteorite volume (Siraj et al.).

Separately, we observe a factor of ~ 2 increase in yield strength during the experiment, which we attribute to proton-beam-induced strain hardening. This is consistent with the known relation between dislocation density and yield strength, where yield strength scales with the square root of dislocation density. We have clarified this relation in more detail in the Methods section.

To this end, we explored the possibility of conducting measurements at a large-scale neutron diffraction facility, with the objective of comparing a reference sample from the same Campo del Cielo meteorite block (unirradiated) to its counterpart irradiated at CERN's HiRadMat facility. However, due to the activation level of the irradiated sample and the stringent precision requirements, the necessary beamline modifications are not yet available. Consequently, such measurements are unfortunately not feasible at this time.

For planetary defense simulations, the bulk yield strength is typically the more appropriate input parameter, as it governs the response of the entire heterogeneous body under dynamic loading. The observed factor-of-two increase in this bulk property suggests that irradiation-induced hardening may enhance the energy absorption capacity of asteroid material during a deflection event.

- 6) *Some discussion would be useful on how these results extend to other planetary defense methods. As of now, proton beam irradiation is not one of the common techniques for asteroid deflection. Does this radiation induced hardening extend to other forms of radiation – such as that produced from a nuclear explosive device – or is it unique to GeV protons? Further, is this unique to iron-nickel meteorites or would you expect to see this hardening in more common rocky asteroid materials?*

Reply: The results of the experiment can be extended both to other forms of radiation and to other forms of asteroid material. We added two paragraphs at the end of the introduction section (lines 95 and the following), summarizing the following:

a) Other forms of radiation

There is no indication that the proton charge played a crucial role. The material behavior was mainly determined by the deep penetration into the material at 440 GeV. Therefore, as long as a comparable high penetration depth is achieved, similar results can be expected with neutrons or gammas.

The output of a nuclear device are mainly soft x-rays, that do not penetrate deeply into the material (Miles, 2008).

In our experiment, the beam focus was relatively small compared to the sample's diameter. The measured oscillations were the result of a thermo-elastic wave in the material which is secondary to the p^+ energy deposition.

This suggests that a sufficiently strong shockwave as a secondary result of a nuclear device could lead to similar results. This is further supported by results from nuclear tests where metal plates survived far beyond the melting point. Asteroid data of collisional shocks show that pressures at 10 GPa and beyond suffice to get comparable results to our experiments. By choosing a suitable combination of the required standoff and corresponding yield that allow for producing even TPa-sized pressures, nuclear devices can lead to a similar behavior.

b) Other asteroid types with other material

The question concerning other asteroid types, i.e. other asteroid materials than iron-nickel is highly relevant since stony asteroids are the most common type of near-Earth asteroids. For in-beam experiments at CERN, material parameters need to be sufficiently known to fulfill experimental safety requirements. Iron-rich meteorite material is more homogeneous than stony meteorite material, which makes it easier to describe the material with single parameters such as yield strength. A reliable safety model – applicable for further in-beam experiments at CERN - for using stony meteorites, would require a 3D model comprising a set of material properties. We are currently developing such a model, which would allow for a proper safety model opening up the possibility of experiments with more inhomogeneous materials. Very generally one would indeed expect that stony objects could show a similar behavior of increasing yield strength since at least similar patterns of phase transitions as compared to metal-rich objects are known for these objects under very high pressures.

Reviewer #2 (Remarks to the Author):

In this study the authors present an experiment conducted at the High-Radiation to Materials (HiRadMat) facility of the European Organization for Nuclear Research (CERN) in which a meteorite sample is exposed to a high intensity of high energy radiation and the dynamical response of momentum transferred to the sample is precisely measured in real-time. An accurate knowledge of the material response of the asteroid is critical in various fields: in particular, in asteroid deflection techniques the object response is to be known very precisely to model deflection orbits accurately and to predict the transfer of kinetic energy. In the experimental Campaign presented in the manuscript, an iron-meteorite sample is, for the first time, irradiated with 440 GeV Protons delivered by the Super Proton Synchrotron.

MAJOR COMMENTS

- 1) *In the tests a Laser Doppler Vibrometer is utilized to measure the thermally induced stress waves in real-time, produced by energy deposition in the sample. This means that only radially polarized Waves are measured, neglecting the contribution to moment transfer due to longitudinal oscillations which may assume non-negligible amplitudes in this sort of tests [1]. If so, please justify this assumption.*

Reply: We highly appreciate this insightful comment. It is indeed correct that only radial oscillations were measured. The primary objective of the Laser Doppler Vibrometry (LDV) measurements was to obtain real-time data on changes in the material's yield strength, which is most sensitively reflected in the radial response. While longitudinal oscillations are indeed coupled to radial ones via the Poisson ratio and are therefore non-negligible, our assumption to focus on the radial component is based on two main considerations:

- A) The Poisson ratio, which governs this coupling, is not expected to change significantly during the beam exposure, in contrast to the yield strength, which is known to vary under these extreme conditions.
- B) The available measurement geometry and access constraints made radial measurements more feasible and reliable for real-time data acquisition.

Nevertheless, we agree that a more complete picture would benefit from capturing both components. In future experiments, we plan to implement complementary diagnostics capable of resolving longitudinal oscillations to further refine the interpretation of stress wave dynamics and momentum transfer.

We added a clarification on this in the Experimental setup section, as well as in the Methods sections.

- 2) *In line 125 it is written that the pulse intensity is chosen so to be isolated from solid-to-solid phase transitions: in the following lines, however, it is said that such transitions are present in Asteroid impacts whose momentum transfer characterization constitutes the primary scope of the study. Please explain.*

Reply: We appreciate the reviewer's observation. The reference to asteroid impacts in the manuscript serves only to illustrate the broader relevance of momentum transfer processes in planetary defense scenarios. However, our study focuses on experiments designed to remain below the threshold for solid-to-solid phase transitions in order to isolate and study specific material response mechanisms under controlled conditions.

The solid-to-solid phase transitions mentioned in the context of asteroid impacts occur at much higher energy densities and serve as an ultimate point of comparison—not as a direct analog to our present test conditions. Our experimental strategy follows a staged approach: the current work focuses on regimes without phase transitions to establish a clear baseline, while future experiments with adjusted beam parameters are planned to explicitly explore phase transition effects.

To prevent any potential misunderstanding, we included this distinction starting at line 242, alongside the explanation already given in the Experimental Setup section.

- 3) *The use of thermal stress models as the one cited in [2] requires either the knowledge of the material properties of the samples or the implementation of numerical simulations to be compared to experimental data for benchmarking and derivation of the said constitutive parameters, as done in [1]. With regard to the latter case, only reference to “calculations” is made in the manuscript (e.g. in line 185). Please explain.*

Reply: Indeed, as correctly noted, the application of thermal stress models requires input material properties such as Young’s modulus, Poisson ratio, and thermal expansion coefficient. These were incorporated into the analytical model in the manuscript, using literature values for Kamacite and Taenite as listed. Since this was not explicitly comprised in the manuscript, we added the corresponding values in table 1.

The "calculations" mentioned in the manuscript were obtained using the semi-analytical approach described in Bertarelli et al. [2], rather than from full numerical simulations as conducted in [1].

Our approach focuses on comparing these analytically calculated stress levels to the yield strength values inferred from the LDV-measured surface vibrations. The observed consistency between the analytically predicted stress thresholds and the experimentally detected transitions — from elastic behavior to plastic deformation, and subsequently to oscillatory behavior indicative of strain hardening — is fully consistent with the selected material parameters.

We acknowledge that a full numerical treatment, as performed in [1], could provide additional insight. Incorporating such simulations is part of our planned future work, especially as we extend the experimental program to more complex geometries and loading conditions.

- 4) *In line 211 it is claimed that an unchanged spectrum of the sample’s response in presence of induced plasticity indicates a very homogeneous hardening process of the material. A homogenous change of the material properties, however, would typically reflect in a substantial change in the frequency content of the acquired response. Please explain why this is not the case.*

Reply: We agree with the reviewer that a homogeneous change in material properties would generally be expected to alter the frequency content of the response. In our case, however, we have no evidence that the Young’s modulus, changed significantly during the experiment. Since the eigenfrequencies primarily depend on the elastic modulus and geometry, we would not expect a substantial shift in the spectrum if the Young’s modulus remains constant.

The observed plastic deformation appears to have manifested primarily as strain hardening, increasing the yield strength, but without significantly affecting the elastic behavior that governs the vibrational spectrum. Furthermore, we found no indication of macroscopic geometric alterations, which would typically induce noticeable changes in the spectrum, especially at higher frequencies.

Thus, the unchanged frequency content supports the interpretation that plasticity was accommodated through homogeneous strain hardening without substantial modification of the elastic properties or geometry that would affect the measured spectrum.

- 5) *In line 222 it is reported that strain hardening and amplitude-dependent damping effects are observed in the experimental response of the sample: please explain how these effects are included in the material model adopted in the study to predict the stress levels indicated.*

Reply: Thank you for this important observation. The thermal stress model by Bertarelli et al. [2], which we used to estimate the stress levels in the sample, is based on linear elasticity and does not incorporate nonlinear material effects such as strain hardening or amplitude-dependent damping. These effects were identified retrospectively through analysis of the LDV data and are therefore not part of the original predictive model.

Following this comment, we revised the Discussion section to more clearly distinguish between the material model used for stress estimation and the experimental observations. Specifically:

- Thermal stress model (Bertarelli et al.): Used to estimate the applied stress, based purely on linear elasticity. This model does not include nonlinear effects such as strain hardening or amplitude-dependent damping. A more detailed description is now provided in the Methods section.

- Strain hardening: Inferred from the comparison between the applied thermal stress and the estimated bulk yield strength. At higher beam intensities, the stress exceeds the yield strength, leading to plastic deformation.
- Plastic deformation energy: Quantified from the suppression of oscillation amplitude in the LDV data. This energy was interpreted as being stored in newly formed dislocations, resulting in an increased dislocation density—and thus an increase in yield strength consistent with strain hardening behavior.
- Amplification factor ($f = 7$): Introduced to reconcile the discrepancy between local (nanoindentation-based) and bulk (fragmentation-based) yield strength values. This factor reflects internal inertial effects and heterogeneous failure behavior within the sample.
- Amplitude-dependent damping: Observed in the high-intensity LDV data and attributed to internal inertial motions due to microstructural features (e.g., inclusions, phase boundaries). This nonlinear effect was not captured by the material model, as it was not anticipated.

These updates clarify the distinction between predictive modeling and experimentally observed nonlinearities, both of which are now treated separately and in more detail both in the Discussion and in the Methods section.

MINOR COMMENTS

6) *How is the implemented number of pulses and the pulse intensity chosen? Please justify.*

The selection of pulse intensities and number of pulses was driven by the objective to compare two well-separated material response regimes. The lower intensity of $1 \times 10^{11} \text{p}^+$ was chosen to keep the material response within the elastic regime, while the higher intensity of $3 \times 10^{11} \text{p}^+$ was selected to deliberately induce a distinguishable onset of plasticity, allowing for direct comparison between elastic and plastic behavior.

Since this was the first experiment of this type performed worldwide on iron meteorite samples under these conditions, the higher intensity was conservatively selected to remain within safety margins, providing sufficient buffer below critical thresholds where phase transitions or structural failure might occur.

The number of shots was chosen to ensure that the total integrated beam energy remained safely below levels that could cause fatal damage to the sample or compromise facility safety, while still allowing the accumulation of sufficient statistics for the analysis of both elastic and plastic response regimes.

We added a new subsection “Pulse intensity and shot number selection” on this in the Methods section.

7) *Please provide a superposed view of the FFT response spectra associated to low/high-intensity pulses shown in Fig. 3b.*

We have replaced the initially provided FFT graphs by a FFT overlay view, showing the dominant frequency components in the 375 – 450 kHz range shortly after beam impact. Further details are included in the subsection “Time-frequency analysis via continuous wavelet transform” under Methods.

References

[1] Pasquali, M., Bertarelli, A., Accettura, C. et al. Dynamic Response of Advanced Materials Impacted by Particle Beams: The MultiMat Experiment. *J. dynamic behavior mater.* 5, 266–295 (2019).

<https://doi.org/10.1007/s40870-019-00210-1>

[2] Bertarelli, A., Dallochio, A., Kurtyka, T.: Dynamic response of rapidly heated cylindrical rods: Longitudinal and flexural behavior. *Journal of Applied Mechanics* 75(3) (2008).

<https://doi.org/10.1115/1.2839901>

Reviewer #3 (Remarks to the Author):

The authors examine the effect of high-energy (440 GeV) proton irradiation on a meteorite sample using the HiRadMat facility at CERN. The subject is of broad scientific interest, and the experiment itself is quite interesting. The authors observed a change in the structural dynamic response of the cylindrical meteorite sample, as measured with a laser doppler vibrometer, most of the time occurring when the beam irradiance was increased. The authors attribute this change in signal to energy loss and assert it arises from plastic deformation. Based on this interpretation, they estimate the amount of strain hardening the sample may have experienced. While the authors reasonably show that this is plausible, it is not convincing without further discussion to eliminate other possibilities and/or without further quantification of uncertainties.

MAJOR COMMENTS

- 1) *The single LDV measurement directed orthogonally to the rod should be primarily sensitive to flexure and/or radial dilation, whereas longitudinal modes should also exist (see Ref.22) yet are not accounted for in the analysis. Rather, all energy is assumed to be contained within the radially directed wave, which may not be strictly correct. In particular, there is no comparison of the wave energy to the beam energy for the so-called “elastic case”, or to some other control, to prove this methodology or to otherwise quantify uncertainties in determining the quantity of energy dissipated through plastic deformation. Similarly, there is no discussion of other modes, including the possibility of surface waves, and if those could produce the change in signal pattern at the higher beam irradiance.*

Reply: We thank the reviewer for raising this important point. The primary objective of the Laser Doppler Vibrometry (LDV) measurements was to obtain real-time data on changes in the material's yield strength under beam exposure, which is most directly and sensitively reflected in the radial response of the cylindrical sample. Particularly, we added further clarification on this in the Experimental setup section, but also tried to account for this by reorganizing the Discussion section.

We fully acknowledge that longitudinal modes also exist (see Ref. 22/ Bertarelli et al.) and are coupled to radial modes via the Poisson ratio. However, our analysis focused on the radial response for the following reasons:

- a) Dominant Sensitivity to Yield Strength Changes:

The radial response provides the most direct signature of changes in yield strength, which is the primary parameter affected by the beam-induced rapid heating and subsequent strain hardening. In contrast, the Poisson ratio governing radial-longitudinal coupling is not expected to vary significantly during irradiation, while the yield strength does.

- b) Experimental constraints and signal quality:

The measurement geometry and experimental access favored radial LDV measurements, which allowed us to obtain high-quality, real-time displacement data with sufficient signal-to-noise ratio for quantitative analysis. Longitudinal measurements would have required a significantly more complex setup, which was beyond the scope of the present study.

- c) Elastic vs. plastic regime:

We primarily compare the oscillation behavior before and after the onset of plastic deformation. In the elastic regime, the sample exhibits stable oscillations. After plastic deformation, these oscillations are damped due to energy dissipation into dislocation formation and movement. Based on our dislocation density calculations, approximately 10 shots are sufficient for the system to reach a new equilibrium, after which stable oscillations reappear — consistent with the experimental observation.

- d) Energy deposition and FLUKA simulations:

We did not explicitly compare the measured mechanical energy to the incident beam energy because, at 440 GeV, the nuclear length of the protons exceeds the sample length (10 cm), and thus the total beam energy is only partially deposited in the sample. The spatial distribution and

magnitude of deposited energy were quantified using FLUKA simulations, which account for both radial and longitudinal energy deposition profiles. In our LDV analysis, the radial energy deposition was weighted according to these FLUKA results. Thus, in the energy loss calculations to arrive at a value for hardening, the longitudinal dependence was indirectly taken into account by weighing the radial data with the energy deposition map of the Monte Carlo simulation.

e) Other modes and surface waves:

We agree that additional modes such as surface waves could, in principle, contribute to the observed signal, particularly at higher beam intensities. However, given the observed spectral stability of the oscillations after multiple shots and the clear correlation with plastic hardening, we interpret the dominant contribution to stem from bulk radial modes. Nevertheless, we fully recognize that capturing a more complete modal picture — including longitudinal and surface waves — would further improve the interpretation. We plan to address this in future experiments by implementing complementary diagnostics capable of resolving these additional components.

- 2) *The observed resonance frequencies (Fig.3b) are not sufficiently explained. For example, if one assumes a nominal wave speed of ~3000 m/s, the primary resonances (Fig.3a) have wavelength ~1.8 m, which does not appear to correspond to a primary geometric feature of the sample. One wonders if this could indicate leakage of mechanical energy to the fixture (“envelope”).*

Reply: We thank the reviewer for this valuable observation. To clarify the origin of the observed resonance frequencies in the LDV data, we now explicitly discuss the relevant geometrical and material parameters in the revised manuscript. Specifically, we consider the fundamental geometrical scale of the sample, defined by its 10 mm diameter, which governs the radial resonance modes.

While a nominal sound speed of 3000 m/s could suggest larger-scale wavelengths, the meteorite’s heterogeneous composition—primarily Kamacite and Taenite phases—leads to a broad range of effective sound velocities. Reported values range from approximately 5500 m/s in Kamacite to 3000 m/s in Taenite (Ueki et al), and the effective propagation paths are influenced by the specific spatial arrangement of these phases. As a result, the resonance conditions reflect complex internal structure rather than a single uniform wave speed.

We have added a paragraph in the Results section clarifying that the dominant frequencies in the FFT spectra (375–450 kHz) are in good agreement with expected fundamental radial modes for the sample’s size and composite sound velocity. Furthermore, the consistent reappearance of these frequencies across multiple shots suggests that the measured response originates from internal volumetric stress waves within the sample, not from coupling to the envelope. We have replaced the figures with an FFT overlay view, showing the dominant frequencies after beam impact.

- 3) *The initial temperature of the rod before each pulse should be described. Was it allowed to equilibrate? Did the rod always start at room temperature, or did it get hotter with each radiation pulse? What is the effect of successive heating? It may also help to describe the time between radiation pulses.*

Reply: We thank the reviewer for pointing this out. The temperature development over time and beam shot was only provided in the supplementary material. We now added further information on the beam irradiation timing and mechanical and thermal equilibration of the sample in the caption of Figure 2. While the temperature did not always return fully to room temperature between pulses, the sample was allowed to equilibrate prior to each beam shot, and the time between consecutive shots was sufficiently long. Due to the nature of the experiment, where very high stress and strain levels dominate the response, these comparatively small variations (of only a few Kelvin) in initial temperature are not expected to have a significant influence on the mechanical response or on the interpretation of the strain hardening effects observed. The typical time between subsequent radiation pulses was on the order of minutes.

- 4) *One would like to see a quantitative measure of dislocation density before and after irradiation, or some other independent measurement, to confirm the author's hypothesis regarding structural change. In general, I do not find the explanation of strain hardening convincing. I agree that this is*

an intriguing possibility with important implications and is worth exploring. However, reaching this conclusion depends on correctly interpreting a very complex series of behavior that occurred sequentially, whereas the experiments only measured one aspect of this behavior or the interpretation may not be correct (see my other comments).

Reply: We fully acknowledge the reviewer's point that an independent quantification of dislocation density would provide valuable confirmation of the hypothesized structural changes. To this end, we explored the possibility of conducting measurements at a large-scale neutron diffraction facility, with the objective of comparing a reference sample from the same Campo del Cielo meteorite block (unirradiated) to its counterpart irradiated at CERN's HiRadMat facility. However, due to the activation level of the irradiated sample and the stringent precision requirements, the necessary beamline modifications are not yet available. Consequently, such measurements are unfortunately not feasible at this time. As discussed in Osuch et al., the initial dislocation density in Kamacite is typically low, making the material highly sensitive to dislocation generation under increasing stress and strain conditions. The general mechanism of strain hardening via dislocation density increase is well-established from related studies. Our current experimental data demonstrate that strain hardening effects can still be inferred on short time scales under the high stress and strain-rate conditions of the irradiation. Prior to our experiments it would still have been possible that the well-known quick hardening of Kamacite under stress might not appear under very high strain rates and the material might not react on these very short time scales. Our results closed this knowledge gap which is of fundamental importance to any nuclear deflection scheme for asteroids.

MINOR COMMENTS

- 5) *The motivation for placing a phase boundary beneath the focal spot of the LDV needs more explanation. In particular, how would a "preferred direction" of the stress wave be determined from this single point measurement? How was it determined that there was no "preferred direction"?*

Reply: Thank you for this thoughtful question. The placement of the LDV sensor above a known phase boundary was a deliberate choice to test whether stress waves induced by high-energy proton heating would exhibit directional propagation effects associated with microstructural inhomogeneities.

In heterogeneous materials like meteorites, even ppm-level inclusions can lead to anisotropic stress wave behavior under dynamic loading, as shown in prior studies (see Jain et al). This can result in preferential wave propagation paths or spectral splitting, especially across phase boundaries. To test whether such effects occurred in our experiment, we placed the LDV sensor directly over a visible Kamacite–Taenite boundary.

A key observation is that no directional or spectral asymmetries were detected in the resulting LDV signal. The Fourier spectra showed consistent peak positions and amplitudes across repeated shots, with no evidence of spectral splitting or shifts. This strongly suggests that the stress field induced by the proton pulse was volumetric and spatially uniform — in stark contrast to impact-driven shock waves, which are known to exhibit anisotropic propagation along microstructural features.

Our decision was further motivated by previous observations that structural transitions in iron-rich meteorites (e.g., formation of the ϵ -phase) tend to initiate at phase boundaries, particularly under high-pressure impact conditions (see e.g. Jain et al or Osuch et al). While our experiment involves thermal stress rather than mechanical shock, we hypothesized that the boundary could still act as a sensitive probe for inhomogeneous stress propagation — which, however, was not observed. This supports the assumption of a largely homogeneous thermoelastic field in our experimental setup.

In the revised version of the manuscript, we sharpened the focus on the crucial effects of hardening and strain-amplitude dependent damping.

- 6) *Fig.1b: Scale bar does not exist or is too small to see, which makes interpretation of the image difficult.*

ReplyThe reviewer’s comment revealed that Figure 1 was not labeled correctly. In response, we have divided the figure into two panels: (a) on the left-hand side and (b) on the right-hand side. The scale bar refers to panel (a), while panel (b) illustrates the untreated and cut samples of the Campo del Cielo meteorite.

7) *Fig.2: There is no discussion of the shots with “no clear response”. Does this imply difficulties with the measurement? What is the implication for accuracy of the other shots?*

Reply: The shots with “no clear response” refer to beam shots #24 and #27. These were not included in the aggregation of the main groups, as their displacement profiles exhibit a significantly higher standard deviation compared to the other measurements. Out of the total 27 shots, 2 were used for calibration, 2 exhibited no clear response, and 23 could be reliably classified as showing either elastic or plastic oscillatory profiles. Therefore, the presence of these two outlier shots does not affect the accuracy or reliability of the classification and analysis of the remaining data.

To further support this, we updated Figure 3a to include the standard deviation for each of the three main groups of LDV response profiles. The resulting error bars demonstrate that the grouped data are highly consistent, with only small intra-group variation.

This clarification has been added to the revised Results section and is now reflected in the caption of Figure 3.

8) *Fig.2: Authors should check the units in the energy scale; they do not appear consistent with the energy deposition values in Fig.1a.*

Reply: Thank you for pointing this out. We confirm that the energy units in Figures 1a and 2 are consistent, though they represent different quantities:

- Figure 2 shows the energy deposition density per primary proton, given in units of GeV/cm³ per proton.
- Figure 1a shows the total energy deposition density, given in J/cm³ for a full beam shot.

To illustrate the consistency:

- A value of 5.9 GeV/cm³ per proton in Fig. 2 corresponds to:
 $5.9 \text{ GeV/cm}^3 \times 1.602 \times 10^{-10} \text{ J/GeV} = 9.45 \times 10^{-10} \text{ J/cm}^3 \text{ per proton}$
- For a shot with 3×10^{11} protons:
 $9.45 \times 10^{-10} \text{ J/cm}^3 \times 3 \times 10^{11} = 283 \text{ J/cm}^3$

This value is fully consistent with the upper end of the color scale in Figure 1a, which tops out at approximately 300 J/cm³.

We have added a clarification to the revised figure captions to avoid confusion.

9) *Line 119—122: This sentence appears overstated, as the present study does not show how proton beams could be used to deflect or maneuver asteroids. The present study only addresses the wave behavior.*

Reply: Thank you for this important clarification. We fully agree that the present study does not directly demonstrate the feasibility of asteroid deflection or maneuvering using proton beams. Rather, our goal is to address a critical gap in the understanding of material behavior under extreme energy deposition — a key prerequisite for modeling such scenarios realistically.

Previous simulation studies, including those by Moore et al. and the Lawrence Livermore National Laboratory team, have relied on static or assumed material parameters when modeling high-energy deflection concepts. Our work provides, to our knowledge, the first real-time experimental data capturing dynamic changes in yield strength and mechanical response during high-energy proton irradiation of asteroid material.

Moreover, the study helps resolve a long-standing discrepancy between laboratory-based yield strength measurements (e.g., from nanoindentation) and bulk strength estimates derived from atmospheric meteorite breakup modeling. By quantifying a consistent scaling factor between local and bulk yield strength, we contribute relevant input data for future, more accurate planetary defense simulations.

We appreciate the remark and have adjusted the manuscript accordingly by explaining the relation to planetary defense and asteroid deflection scenarios more clearly (Introduction and Experimental setup section).

10) Fig.3a: The two right-most of the three graphical panels have the same labels. Consider differentiating them. Also, it is unclear why the response returns to “elastic” on shot 23 even though the beam energy remains high.

Reply: We thank the reviewer for this comment. Based on this, we restructured the manuscript to clarify the material response effects and streamlined nomenclatures to avoid any misunderstandings.

The goal of the three panels in figure 3a) was to show the exemplary progression of the material response behavior. Starting with low intensity beam shots at 1×10^{11} protons an oscillatory behavior is observed, then beam intensity is increased to 3×10^{11} protons, resulting in non-oscillatory behavior and finally the oscillatory behavior returns at 3×10^{11} protons.

The return of oscillatory behavior at high beam intensity — following a regime of non-oscillatory displacement — indicates that the material experienced a strain-induced hardening process, rather than reaching a hardening saturation or failure threshold. As shown in Figure 2 and Figure 3, the collapse of oscillations at beam intensities of 3×10^{11} protons corresponds to the onset of plastic deformation, while the subsequent recovery of oscillations reflects a dynamic increase in yield strength resulting from dislocation buildup in the Kamacite phase.

In other words, the sample initially softens due to plastic deformation, but then hardens progressively as dislocations accumulate. This increases the local and bulk yield strength such that the same high thermal stress (up to ~ 120 MPa) no longer exceeds the (now higher) yield threshold, allowing the sample to sustain elastic oscillations again.

This interpretation is supported by the following:

- Dislocation density increased by a factor of ~ 6 , as estimated from energy deposition and dislocation line energy.
- Yield strength scales with the square root of dislocation density, implying an increase of local yield strength from ~ 350 MPa to ~ 875 MPa.
- Applying the same factor to the bulk yield strength yields ~ 125 MPa, which is just above the peak thermal stress — consistent with the observed reappearance of oscillations.
- The LDV spectra before and after hardening show no sign of damage or anisotropic wave propagation, suggesting structural integrity and uniform response.
- GEANT4/COMSOL simulations also require a correction factor consistent with this hardening behavior to match observed displacements.

To address this more clearly, we reorganized the manuscript to provide a more structured explanation of the plastic regime, the onset of hardening, and the interplay between the different effects involved. We have now clarified in the Discussion section that the observed recovery of oscillations reflects a strain-hardening effect reaching a new dynamic threshold — not an indication of complete hardening or saturation.

Also, we added a note in Figure 3 that vertical scaling differs between graphs for visual clarity.

11) Line 178: “peak-induced stress” is not clearly defined.

Reply: We thank the reviewer for this remark. To avoid misunderstandings we changed the nomenclature from “peak-induced stress” and clarified the definition in the Methods sections. “Peak-induced stress” is equivalent to von Mises stress, i.e.

$$\sigma_{vm} = \frac{1}{\sqrt{2}} [(\sigma_z - \sigma_\theta)^2 + (\sigma_\theta - \sigma_r)^2 + (\sigma_r - \sigma_z)^2], \text{ with}$$

σ_z is axial stress, σ_θ is tangential stress and σ_r is radial stress.

12) Line 189: “internal dynamics” is not clearly defined.

Reply: Thank you for this helpful remark. Indeed, the term “internal dynamics” was not sufficiently defined in the manuscript. The key observation underlying this term is that yield strength measurements of iron meteorite material differ by a heuristic factor depending on the measurement method. Specifically, nanoindentation measurements yield higher values for the yield strength than estimates derived from ram pressure during meteorite explosions in the atmosphere. One hypothesis is that the heuristic factor of approximately ~ 7 between these two

measurements may be attributed to internal inertial movements within the heterogeneous material structure, which we have summarized under the term “internal dynamics.”

Currently, this heuristic factor is not explained by an integrated mechanical model of the material. However, following the reviewer’s valuable suggestion, we conducted additional Monte Carlo simulations using the GEANT4/COMSOL code. These simulations provided a high-resolution temperature profile of the entire meteorite cylinder volume, including the surface region where the temperature was measured via PT100 sensors.

Based on the simulated surface temperature, we estimated the corresponding thermal strain using the relation:

Strain = $\alpha \times \Delta T$ where α is the average coefficient of thermal expansion (20–100°C) and ΔT is the temperature increase at the surface. When comparing the resulting strain derived from the GEANT4/COMSOL simulations to the measured displacement obtained from LDV, we observe a factor of approximately ~5 between them. While no quantitative model yet exists to fully describe this effect, our simulations qualitatively support the hypothesis that internal inertial movements — linked to the microstructural complexity of the material — contribute to this discrepancy, which we have referred to as “internal dynamics”.

The manuscript structure was reorganized and sharpened to improve clarity and emphasize the distinctions between these effects.

- 13) *Lines 166-167, 178-180, 196-205: Consistency with ram conditions or shock impact conditions is not necessarily expected, as yield strengths are known to be sensitive strain-rate and/or shear conditions.*

Reply: We fully agree with the reviewer that consistency with ram or shock impact conditions is not necessarily expected, as yield strengths are indeed highly sensitive to strain rate and shear conditions. The large discrepancy between yield strength values obtained from different measurement methods — namely, nanoindentation versus atmospheric entry (ram pressure) estimates — points to a factor of ~7 difference. Generally, this discrepancy is interpreted as being indicative of the specific internal structure and microstructural complexity of the meteorite material.

To better understand this, we have included a new paragraph in the revised manuscript discussing additional GEANT4 Monte Carlo simulations that were performed following the reviewer’s valuable comments. These simulations allowed us to obtain high-resolution temperature profiles of the meteorite cylinder and, by applying the relation Strain = $\alpha \times \Delta T$, to estimate surface displacements. The comparison of simulated and experimentally measured displacements supports the hypothesis that internal inertial movements within the microstructure contribute to the observed damping behavior and to the measured discrepancies. With a coefficient of thermal expansion of $11.1 \times 10^{-6} \text{ K}^{-1}$ and a sample radius of 5 mm, we derive a displacement on the order of 0.6 μm . The measured average displacement averaged over oscillating high intensity diagrams is 2.9 μm . The discrepancy of a factor of ~5 is on the order of the factor observed between ram pressure-derived values and nanoindentation measurements. This supports the view that the sample as a whole – when irradiated by the proton beam – experiences an effective stress which includes the effect of internal inertial movements.

This conclusion is further supported by the observed damping behavior at high intensity. The observed strain amplitude-dependent super-exponential damping is usually also explained as excitation of internal inertial degrees of freedom in complex materials.

Note that the suggestion that meteorite material – due to the inclusions – behaves like a composite has already been made in Mulford et al.

The structure of the manuscript was reorganized to facilitate a clearer distinction between the effects and improve overall comprehensibility.

- 14) *Fig.5: The meaning of the color bar is unclear, specifically the normalization “per bin of 1 cm³” is sometimes used for energy but not usually for temperature. Should the units instead be “K/pulse? Or this normalization should be described.*

Reply: Thank you for this helpful comment. Figure 1 and Figure 5 are based on two separate Monte Carlo simulations. Figure 1 presents the simulated local energy deposition profile in

J/cm³, while Figure 5 shows the corresponding local temperature increase derived from a dedicated simulation focusing on temperature development. Based on the reviewer's comment, we conducted an additional Monte Carlo simulation using COMSOL and GEANT4 with higher spatial resolution over the full meteorite cylinder volume to obtain a more detailed temperature development. In the revised manuscript, we replaced Fig. 5 with the new high-resolution GEANT4/COMSOL simulation, which now provides a more precise spatial representation of the local temperature increase. The color scale indicates the temperature increase per pulse, in units of K/pulse. We updated the figure caption and methods section accordingly to clarify this normalization. The main purpose remains to demonstrate that, even at the highest beam intensities, the temperature increase remains far below phase transition thresholds of the material.

15) Line 301-303: *It does not seem correct to say “the PT100 sensor measured roughly the same amount of temperature increase as for the oscillating diagrams” because the diagrams do not measure temperature. Or the method behind this inference should be explained more fully.*

Reply: Thank you for this remark. We agree that the original wording was not precise, as the oscillatory diagrams obtained by Laser Doppler Vibrometry do not measure temperature directly. We have therefore revised the sentence to: “The PT100 sensor measured roughly the same amount of temperature increase as for the beam shots characterized by oscillatory behavior.” This wording more accurately reflects that the comparison refers to beam shots associated with oscillatory response, not to temperature measurements derived from the LDV data itself.

16) Line 324-325: *Why was this weighting performed? Does it imply a systematic error in the model?*

Reply: Thank you for this important question. We added a more structured explanation in the Methods section capturing the following: The weighting was performed because a simple homogeneous integration of the measurement data, taken at $z = 5$ cm (middle of the cylinder), would not properly reflect the fact that the energy deposition is strongly position-dependent along the longitudinal (z -) axis of the cylinder, as shown by the FLUKA simulation. The weighting compensates for this effect by scaling the LDV-derived local energy with the ratio of simulated temperature at each position to the PT100 reference, thus providing a more accurate estimate of the total energy deposited into plastic deformation across the entire sample volume. Of course, performing LDV measurements at multiple longitudinal positions would be preferable, but this was not feasible due to experimental limitations at CERN related to equipment access restrictions in high-radiation zones.

17) Line 329-330: *The values (with units) when used in (2) do not appear to give units of energy; they give units of force. Also, it is unclear how the dislocation density was calculated from (2).*

Reply: First, we estimated the energy loss in the oscillatory LDV signals by comparing the energy content during oscillatory, i.e. elastic (low-amplitude) and non-oscillatory, i.e. plastic (suppressed amplitude) phases, summing across all beam shots. The total plastic deformation energy was determined to be $E_{\text{plastic}} = 360 \mu\text{J}$. This value was corrected for spatial distribution using Monte Carlo thermal simulations (FLUKA), normalized to the PT100 temperature sensor data.

Assuming that this energy is fully stored in newly formed dislocations, we used the relation $E_{\text{disloc}} = \frac{1}{2} G b^2$ with $G=75$ GPa, and $b = 1.44 \text{ \AA}$ to obtain a dislocation line energy of $7.78 \times 10^{-10} \text{ J/m}$. This corresponds to a calculated dislocation density increase of $6.1 \times 10^{10} \text{ m}^{-2}$, consistent with prior studies on Kamacite which show that significant hardening can occur at relatively low increases in dislocation density compared to typical metals (Osuch et al).

Following the empirical relation between yield strength and dislocation density (approximately the square root of the dislocation density), this would translate into a hardening of Kamacite from an initial yield strength of ~ 350 MPa (as obtained by nanoindentation measurements \cite{ueki2021excellent}) to ~ 875 MPa. Again, based on this comment and others addressing the hardening process and underlying calculations, we reorganized the manuscript, particularly in the Discussion and Methods section.

17 October 2025

Dear Reviewers,

We would like to thank you for taking the time to consider the revised version of our manuscript entitled “*Dynamical development of strength and stability of asteroid material under 440 GeV proton beam irradiation*” (Manuscript ID: NCOMMS-24-76790). We are grateful for your detailed and valuable feedback, which has significantly helped us to strengthen and clarify our study.

In this second revision, we have addressed the remaining points raised in the previous round and included additional work to further consolidate our findings. The main updates concern:

- Application of the Grüneisen parameter to derive local pressure values from the higher-resolution temperature profile simulations, offering an independent and quantitative consistency check for our interpretation of the observed material response.
- Expanded discussion of these results taking into account more recent work on high-pressure phase behavior of Fe–Ni alloys and a comparison between the Bertarelli thermal-stress model and the local peak pressures derived via the Grüneisen approach.
- Explicit clarification that the LDV measurements represent volumetrically averaged responses.
- Besides we modified wording, replaced or added relevant references and polished two figures for better clarity.

In addition to the revised manuscript, we have uploaded a version with all textual changes highlighted in blue (figures omitted for improved readability). We believe that these additions further strengthen the robustness of our conclusions and improve the overall clarity and readability of the paper.

Thank you again for your constructive evaluation and consideration of our work.

Sincerely,

Melanie Bochmann (on the behalf of all co-authors)

Reviewer #1 (Remarks to the Author):

The edits and thoughtful responses provided by the authors have addressed all of my comments/concerns about the paper from the first revision. There are some very minor issues that remain that I have detailed below:

- 1) *Citation 1 (Miles et al) is probably not the best citation for general Planetary Defense, including references to Z-machine experiments. I would suggest either citing the Dearborn and Miller for a general overview of Planetary Defense and early work by Remo on Z-machine experiments.*

Reply: Thank you for this suggestion. We fully agree and have cited Dearborn & Miller for the general overview of Planetary Defense in the first paragraph of the introduction. In addition, we included Remo's early Z-machine studies as a more relevant reference for laboratory-scale simulations.

- 2) *Line 176 has a colon like it is going to provide a definition - I think it should be a period.*

Reply: Thank you for noting this. We agree — the colon at the end of line 176 was unintended and was replaced by a period in the revised version.

Reviewer #2 (Remarks to the Author):

All the comments provided by this reviewer have been carefully addressed and fully incorporated, leading to a substantial improvement in the overall quality of the manuscript. Therefore, the article is now deemed suitable for publication.

Reviewer #3 (Remarks to the Author):

The authors have substantially improved both the rigor and clarity of this work. Several issues still remain as outlined below.

MAJOR COMMENTS

- 1) *The temperature map (Fig.6), based on revised, higher-resolution simulations, shows substantially higher peak temperatures than before. Because the heating is nearly instantaneous (250 ps per pulse), one can use the Grüneisen parameter to estimate local pressure of several GPa in the region of maximum energy deposition, which is an order-of-magnitude higher than the peak stress estimated by the authors using the method of Bertarelli based on the response of the entire rod. This reintroduces the question of localized, solid-solid phase transition contributing to the energy dissipation. The authors cite two papers on the high-pressure Fe-Ni alloys may undergo fcc to hcp transition under conditions of a few hundred Kelvin temperature rise and a few GPa pressure (as an example only, see Torchio 2020, DOI:10.1029/2020GL088169). Given present uncertainties in phase behavior in this regime, more discussion or analysis on this point seems necessary to distinguish between solid-solid phase transition and the proposed defect production.*

Reply: We thank the reviewer for the valuable suggestion to use the Grüneisen approach for estimating the instantaneous thermal pressure under ultrafast (≈ 250 ps) isochoric heating.

We calculated the Grüneisen parameter γ using the relationship $\gamma = \frac{\alpha_V \cdot K_T}{\rho \cdot c_V}$, using the thermophysical parameters for Fe-Ni in the dominating kamacite phase:

Property	Value	Unit	Reference
Thermal expansivity, α	$11.1 \cdot 10^{-6}$	K^{-1}	[27]
Young's modulus, E	240	GPa	[24]
Poisson ratio, ν	0.3		[28]
Density, ρ	8060	kg/m^3	[18]
Specific heat coefficient, c_p	461	J/kg K	[25,26]
Derived properties			
Bulk modulus, K_T	200	GPa	$K_T = E/(1-2\nu)$
Volumetric thermal expansivity α_v	$11.1 \cdot 10^{-6}$	K^{-1}	$\alpha_v = 3\alpha$

These values yield a Grüneisen parameter of $\gamma \sim 1.8$. Applying $p \approx \gamma \cdot \rho \cdot c_v \cdot \Delta T$ to the peak temperature rises from the revised, higher-resolution simulation yields local thermal pressures between 2.2 and 3.0 GPa (details are in the table below).

$\Delta T_{\text{threshold}}$ [K]	V_{heated} [mm^3]	p_{local} [GPa]
300	4.0897	2.2
350	1.3216	2.5
400	0.1364	2.8
430	0.0056	2.9
435	0.0015	2.9
440	0.0002	2.9

As the reviewer correctly notes, these local pressures are approximately an order of magnitude higher than the meteorite cylinder rod-averaged thermal peak stress using the thermal stress model by Bertarelli et al.

For comparison, Torchio et al. (2020) report that in Fe–Ni alloys the bcc \rightarrow hcp ($\alpha \rightarrow \epsilon$) transition starts at pressures above ≈ 12 GPa for Ni ≈ 20 wt %, and completes at 17 GPa, shifting to even higher pressures (> 100 GPa) with increasing Ni content. The average Ni concentration in our iron meteorite sample is significantly lower (≈ 6.15 wt % overall, ≈ 5.5 wt % for the dominant bcc kamacite phase). Consequently, the 2–3 GPa local pressures estimated here lie well below any known equilibrium phase-transition thresholds, and we therefore do not expect solid–solid phase transitions under these conditions.

We have included this additional consistency check in the summary part of the Discussion section (line 191) and added a dedicated paragraph in the Methods section (line 363) providing further details of the calculations.

- 2) *Following the above, the authors should consider if the response measured with LDV should be contextualized as a volumetrically-averaged result or similar, and the potential influence (or not) of localized heating and stress may need to be discussed.*

Reply: Correct, LDV is only volumetrically averaged result as it is discussed, will be included explicitly in the manuscript. The localized hot zone ($\Delta T \geq 300$ K) determined from the revised simulation corresponds to only **~0.7 %** of the total cylinder volume, while the thermal stress wave generated there propagates throughout the full sample. An additional explanatory sentence and disclaimer have been included in the corresponding Methods section to clarify this interpretation (line 389).

MINOR COMMENTS

- 1) *Fig. 1b is still missing a scale bar for the inset at upper right, which is necessary for interpreting the image.*

Reply: Thank you for pointing this out. We have replaced the previous image with a simplified top-view photograph to avoid perspective distortion. The new version includes explicit x- and y-dimensions in the caption (21.4 cm × 14.3 cm), which provides a clear spatial reference.

This photograph was added.

- 2) *Abstract, first sentence: “dynamic response” is vague, which makes the statement “an effect that has not been captured until now” not particularly meaningful.*

Reply: We agree that “dynamic response” was too general. We have revised the opening sentence to specify the physical process investigated, clarifying the novelty of the experimental observation. In addition, we modified other text sections accordingly (e.g. in the introduction).

- 3) *Fig.6, fonts are very small and difficult to read on the color bar and elsewhere.*

Reply: Thank you for noting this. We have increased the overall figure size and adjusted the font settings, including those on the color bar, to ensure readability in the revised version.

- 4) *Line 293: As a point of style only, “worldwide” seems overly emphatic and unnecessary.*

Reply: Thank you for this stylistic suggestion. We agree that “worldwide” is unnecessary and have removed it in the revised version for a more concise phrasing.

Review of the article entitled:

“Dynamical development of strength and stability of asteroid material under 440 GeV proton beam irradiation”

by M. Bochmann, K.-G. Schlesinger, C.D. Arrowsmith et Al.

submitted to *Nature Communications*, manuscript ID: NCOMMS-24-76790

In this study the authors present an experiment conducted at the High-Radiation to Materials (HiRadMat) facility of the European Organization for Nuclear Research (CERN) in which a meteorite sample is exposed to a high intensity of high energy radiation and the dynamical response of momentum transferred to the sample is precisely measured in real-time. An accurate knowledge of the material response of the asteroid is critical in various fields: in particular, in asteroid deflection techniques the object response is to be known very precisely to model deflection orbits accurately and to predict the transfer of kinetic energy. In the experimental campaign presented in the manuscript, an iron-meteorite sample is, for the first time, irradiated with 440 GeV protons delivered by the Super Proton Synchrotron.

Major comments

- 1) In the tests a Laser Doppler Vibrometer is utilized to measure the thermally induced stress waves in real-time, produced by energy deposition in the sample. This means that only radially polarized waves are measured, neglecting the contribution to moment transfer due to longitudinal oscillations which may assume non-negligible amplitudes in this sort of tests [1]. If so, please justify this assumption.
- 2) In line 125 it is written that the pulse intensity is chosen so to be isolated from solid-to-solid phase transitions: in the following lines, however, it is said that such transitions are present in asteroid impacts whose momentum transfer characterization constitutes the primary scope of the study. Please explain.
- 3) The use of thermal stress models as the one cited in [2] requires either the knowledge of the material properties of the samples or the implementation of numerical simulations to be compared to experimental data for benchmarking and derivation of the said constitutive parameters, as done in [1]. With regard to the latter case, only reference to “calculations” is made in the manuscript (e.g. in line 185). Please explain.
- 4) In line 211 it is claimed that an unchanged spectrum of the sample’s response in presence of induced plasticity indicates a very homogeneous hardening process of the material. An homogenous change of the material properties, however, would typically reflect in a substantial change in the frequency content of the acquired response. Please explain why this is not the case.
- 5) In line 222 it is reported that strain hardening and amplitude-dependent damping effects are observed in the experimental response of the sample: please explain how these effects are included in the material model adopted in the study to predict the stress levels indicated.

Minor comments

- 1) How is the implemented number of pulses and the pulse intensity chosen? Please justify.
- 2) Please provide a superposed view of the FFT response spectra associated to low/high-intensity pulses shown in Fig. 3b.

References

- [1] Pasquali, M., Bertarelli, A., Accettura, C. et al. Dynamic Response of Advanced Materials Impacted by Particle Beams: The MultiMat Experiment. *J. dynamic behavior mater.* 5, 266–295 (2019). <https://doi.org/10.1007/s40870-019-00210-1>
- [2] Bertarelli, A., Dallochio, A., Kurtyka, T.: Dynamic response of rapidly heated cylindrical rods: Longitudinal and flexural behavior. *Journal of Applied Mechanics* 75(3) (2008). <https://doi.org/10.1115/1.2839901>